# Distributionally Robust Parametric Maximum Likelihood Estimation

**Viet Anh Nguyen**      **Xuhui Zhang**      **José Blanchet**
Stanford University, United States
{viet-anh.nguyen, xuhui.zhang, jose.blanchet}@stanford.edu

**Angelos Georghiou**
University of Cyprus, Cyprus
georghiou.angelos@ucy.ac.cy

## Abstract

We consider the parameter estimation problem of a probabilistic generative model prescribed using a natural exponential family of distributions. For this problem, the typical maximum likelihood estimator usually overfits under limited training sample size, is sensitive to noise and may perform poorly on downstream predictive tasks. To mitigate these issues, we propose a distributionally robust maximum likelihood estimator that minimizes the worst-case expected log-loss uniformly over a parametric Kullback-Leibler ball around a parametric nominal distribution. Leveraging the analytical expression of the Kullback-Leibler divergence between two distributions in the same natural exponential family, we show that the min-max estimation problem is tractable in a broad setting, including the robust training of generalized linear models. Our novel robust estimator also enjoys statistical consistency and delivers promising empirical results in both regression and classification tasks.

## 1   Introduction

We are interested in the relationship between a response variable $Y$ and a covariate $X$ governed by the generative model

$$Y|X = x \sim f\big( \cdot \, |\lambda(w_0, x)\big), \tag{1}$$

where $\lambda$ is a pre-determined function that maps the weight $w_0$ and the covariate $X$ to the parameter of the conditional distribution of $Y$ given $X$. The weight $w_0$ is unknown and is the main quantity of interest to be estimated. Throughout this paper, we assume that the distribution $f$ belongs to the exponential family of distributions. Given a ground measure $\nu$ on $\mathcal{Y}$, the exponential family is characterized by the density function

$$f(y|\theta) = h(y) \exp \big( \langle \theta, T(y) \rangle - \Psi(\theta) \big)$$

with respect to $\nu$, where $\langle \cdot, \cdot \rangle$ denotes the inner product, $\theta$ is the natural parameters, $\Psi$ is the log-partition function and $T$ is the sufficient statistics. The space of natural parameters is denoted by $\Theta = \big\{\theta : \int h(y) \exp(\langle \theta, T(y) \rangle) < \infty \big\} \subseteq \mathbb{R}^p$. We assume that the exponential family of distributions is regular, hence $\Theta$ is an open set, and $T_1(y), \ldots, T_p(y)$ are affinely independent [3, Chapter 8].

The generative setting (1) encapsulates numerous models which are suitable for regression and classification [11]. It ranges from logistic regression for classification [18], Poisson counting regression [17], log-linear models [9] to numerous other generalized linear models [11].

Given data $\{(\widehat{x}_i, \widehat{y}_i)\}_{i=1,\ldots,N}$ which are assumed to be independently and identically distributed (i.i.d.) following the generative model (1), we want to estimate the true value of $w_0$ that dictates (1). If we use $\widehat{\mathbb{P}}^{\mathrm{emp}} = N^{-1} \sum_{i=1}^{N} \delta_{(\widehat{x}_i, \widehat{y}_i)}$ to denote the empirical distribution supported on the training data, and define $\ell_\lambda$ as the log-loss function with the parameter mapping $\lambda$

$$\ell_\lambda(x, y, w) = \Psi(\lambda(w, x)) - \langle T(y), \lambda(w, x) \rangle, \tag{2}$$

then the maximum likelihood estimation (MLE) produces an estimate $w_{MLE}$ by solving the following two equivalent optimization problems

$$w_{MLE} = \arg\min_{w \in \mathcal{W}} \sum_{i=1}^{N} \frac{1}{N} \left( \Psi(\lambda(w, \widehat{x}_i)) - \langle T(\widehat{y}_i), \lambda(w, \widehat{x}_i) \rangle \right) \tag{3a}$$

$$= \arg\min_{w \in \mathcal{W}} \mathbb{E}_{\widehat{\mathbb{P}}^{\mathrm{emp}}}[\ell_\lambda(X, Y, w)]. \tag{3b}$$

The popularity of MLE can be attributed to its consistency, asymptotic normality and efficiency [35, Section 5]. Unfortunately, this estimator exhibits several drawbacks in the finite sample regime, or when the data carry high noise and may be corrupted. For example, the ML estimator for the Gaussian model recovers the sample mean, which is notoriously susceptible to outliers [28]. The MLE for multinomial logistic regression yields over-fitted models for small and medium sized data [10].

Various strategies can be utilized to counter these adverse effects of the MLE in the limited data regime. The most common approach is to add a convex penalty term such as a 1-norm or 2-norm of $w$ into the objective function of problem (3a) to obtain different regularization effects, see [27, 22] for regularized logistic regression. However, this approach relies on strong prior assumptions, such as the sparsity of $w_0$ for the 1-norm regularization, which may rarely hold in reality. Recently, dropout training has been used to prevent overfit and improve the generalization of the MLE [32, 36, 37]. Specific instances of dropout have been shown to be equivalent to a 2-norm regularization upon a suitable transformation of the inputs [36, Section 4]. Another popular strategy to regularize problem (3a) is by reweighting the samples instead of using a constant weight $1/N$ when calculating the loss. This approach is most popular in the name of weighted least-squares, which is a special instance of MLE problem under the Gaussian assumption with heteroscedastic noises.

Distributionally robust optimization (DRO) is an emerging scheme aiming to improve the out-of-sample performance of the statistical estimator, whereby the objective function of problem (3b) is minimized with respect to the most adverse distribution $\mathbb{Q}$ in some ambiguity set. The DRO framework has produced many interesting regularization effects. If the ambiguity set is defined using the Kullback-Leibler (KL) divergence, then we can recover an adversarial reweighting scheme [23, 6], a variance regularization [25, 14], and adaptive gradient boosting [8]. DRO models using the KL divergence is also gaining recent attraction in many machine learning learning tasks [15, 31] and in data-driven optimization [4, 34]. Another popular choice is the Wasserstein distance function which has been shown to have strong connections to regularization [30, 21], and has been used in training robust logistic regression classifiers [29, 7]. Alternatively, the robust statistics literature also consider the robustification of the MLE problem, for example, to estimate a robust location parameter [20].

Existing efforts using DRO typically ignore, or have serious difficulties in exploiting, the available information regarding the generative model (1). While existing approaches using the Kullback-Leibler ball around the empirical distribution completely ignore the possibility of perturbing the conditional distribution, the Wasserstein approach faces the challenge of elicitating a sensible ground metric on the response variables. For a concrete example, if we consider the Poisson regression application, then $Y$ admits values in the space of natural numbers $\mathbb{N}$, and deriving a global metric on $\mathbb{N}$ that carries meaningful local information is nearly impossible because one unit of perturbation of an observation with $\widehat{y}_i = 1$ does not carry the same amount of information as perturbing $\widehat{y}_i = 1000$. The drawbacks of the existing methods behoove us to investigate a novel DRO approach that can incorporate the available information on the generative model in a systematic way.

**Contributions.** We propose the following *distributionally robust MLE problem*

$$\min_{w \in \mathcal{W}} \max_{\mathbb{Q} \in \mathbb{B}(\widehat{\mathbb{P}})} \mathbb{E}_{\mathbb{Q}}\left[ \ell_\lambda(X, Y, w) \right], \tag{4}$$

which is a robustification of the MLE problem (3b) for generative models governed by an exponential family of distributions.

The novelty in our approach can be summarized as follows.

- We advocate a new nominal distribution which is calibrated to reflect the available parametric information, and introduce a Kullback-Leibler ambiguity set that allows perturbations on both the marginal distribution of the covariate and the conditional distributions of the response.

- We show that the min-max estimation problem (4) can be reformulated as a single finite-dimensional minimization problem. Moreover, this reformulation is a convex optimization problem in broadly applicable settings, including the training of many generalized linear models.

- We demonstrate that our approach can recover the adversarial reweighting scheme as a special case, and it is connected to the variance regularization surrogate. Further, we prove that our estimator is consistent and provide insights on the practical tuning of the parameters of the ambiguity set. We also shed light on the most adverse distribution in the ambiguity set that incurs the extremal loss for any estimate of the statistician.

**Technical notations.** The variables $(X, Y)$ admit values in $\mathcal{X} \times \mathcal{Y} \subseteq \mathbb{R}^n \times \mathbb{R}^m$, and $\mathcal{W}$ is a finite-dimensional set. The mapping $\lambda : \mathcal{W} \times \mathcal{X} \to \Theta \subseteq \mathbb{R}^p$ is jointly continuous, and $\langle \cdot, \cdot \rangle$ denotes the inner product in $\mathbb{R}^p$. For any set $\mathcal{S}$, $\mathcal{M}(\mathcal{S})$ is the space of all probability measures with support on $\mathcal{S}$. We use $\xrightarrow{p.}$ to denote convergence in probability, and $\xrightarrow{d.}$ to denote convergence in distribution. All proofs are relegated to the appendix.

## 2 Distributionally Robust Estimation with a Parametric Ambiguity Set

We delineate in this section the ingredients of our distributionally robust MLE using parametric ambiguity set. Since the log-loss function is pre-determined, we focus solely on eliciting a nominal probability measure and the neighborhood surrounding it, which will serve as the ambiguity set.

While the typical empirical measure $\widehat{\mathbb{P}}^{\mathrm{emp}}$ may appear at first as an attractive option for the nominal measure, $\widehat{\mathbb{P}}^{\mathrm{emp}}$ does not reflect the parametric nature of the conditional measure of $Y$ given $X$. Consequently, to robustify the MLE model, we need a novel construction of the nominal distribution $\widehat{\mathbb{P}}$.

Before proceeding, we assume w.l.o.g. that the dataset $\{(\widehat{x}_i, \widehat{y}_i)\}_{i=1,\ldots,N}$ consists of $C \leq N$ distinct observations of $X$, each value is denoted by $\widehat{x}_c$ for $c = 1, \ldots, C$, and the number of observations with the same covariate value $\widehat{x}_c$ is denoted by $N_c$. This regrouping of the data by $\widehat{x}_c$ typically enhances the statistical power of estimating the distribution conditional on the event $X = \widehat{x}_c$.

We posit the following *parametric nominal distribution* $\widehat{\mathbb{P}} \in \mathcal{M}(\mathcal{X} \times \mathcal{Y})$. This distribution is fully characterized by $(p + 1)C$ parameters: a probability vector $\widehat{p} \in \mathbb{R}_+^C$ whose elements sum up to 1 and a vector of nominal natural parameters $\widehat{\theta} \in \Theta^C \subseteq (\mathbb{R}^p)^C$. Mathematically, $\widehat{\mathbb{P}}$ satisfies

$$\begin{cases} \widehat{\mathbb{P}}(\{\widehat{x}_c\} \times A) = \widehat{\mathbb{P}}_X(\{\widehat{x}_c\})\widehat{\mathbb{P}}_{Y|\widehat{x}_c}(A) & \forall \widehat{x}_c, \forall A \subseteq \mathcal{Y} \text{ measurable} \\ \widehat{\mathbb{P}}_X = \sum_{c=1}^C \widehat{p}_c \delta_{\widehat{x}_c}, \quad \widehat{\mathbb{P}}_{Y|\widehat{x}_c} \sim f(\cdot | \widehat{\theta}_c) \, \forall c. \end{cases} \tag{5}$$

The first equation indicates that the nominal measure $\widehat{\mathbb{P}}$ can be decomposed into a marginal distribution of the covariates $X$ and a collection of conditional measures of $Y$ given $X$ using the definition of the conditional probability measure [33, Theorem 9.2.2]. The second line stipulates that the nominal marginal distribution $\widehat{\mathbb{P}}_X$ of the covariates is a discrete distribution supported on $\widehat{x}_c$, $c = 1, \ldots, C$. Moreover, for each $c$, the nominal conditional distribution of $Y$ given $X = \widehat{x}_c$ is a distribution in the exponential family with parameter $\widehat{\theta}_c$. Notice that the form of $\widehat{\mathbb{P}}$ in (5) is chosen to facilitate the injection of parametric information $\widehat{\theta}_c$ into the nominal distribution, and it is also necessary to tie $\widehat{\mathbb{P}}$ to the MLE problem using the following notion of MLE-compatibility.

**Definition 2.1** (MLE-compatible nominal distribution). A nominal distribution $\widehat{\mathbb{P}}$ of the form (5) is MLE-compatible with respect to the log-loss function $\ell_\lambda$ if the optimal solution $\widehat{w} = \arg\min_{w \in \mathcal{W}} \mathbb{E}_{\widehat{\mathbb{P}}}[\ell_\lambda(X, Y, w)]$ coincides with the estimator $w_{MLE}$ that solves (3a).

Definition 2.1 indicates that $\widehat{\mathbb{P}}$ is compatible for the MLE problem if the MLE solution $w_{MLE}$ is recovered by solving problem (3b) where the expectation is now taken under $\widehat{\mathbb{P}}$. Therefore, MLE-compatibility implies that $\widehat{\mathbb{P}}$ and $\widehat{\mathbb{P}}^{\mathrm{emp}}$ are equivalent in the MLE problem.

The next examples suggest two possible ways of calibrating an MLE-compatible $\widehat{\mathbb{P}}$ of the form (5).

**Example 2.2** (Compatible nominal distribution I). If $\widehat{\mathbb{P}}$ is chosen of the form (5) with $\widehat{p}_c = N_c/N$ and $\widehat{\theta}_c = (\nabla\Psi)^{-1}\big((N_c)^{-1}\sum_{\widehat{x}_i=\widehat{x}_c} T(\widehat{y}_i)\big) \in \Theta$ for all $c$, then $\widehat{\mathbb{P}}$ is MLE-compatible.

**Example 2.3** (Compatible nominal distribution II). If $\widehat{\mathbb{P}}$ is chosen of the form (5) with $\widehat{p}_c = N_c/N$ and $\widehat{\theta}_c = \lambda(w_{MLE}, \widehat{x}_c)$ for all $c$, where $w_{MLE}$ solves (3a), then $\widehat{\mathbb{P}}$ is MLE-compatible.

We now detail the choice of the dissimilarity measure which is used to construct the neighborhood surrounding the nominal measure $\widehat{\mathbb{P}}$. For this, we will use the Kullback-Leiber divergence.

**Definition 2.4** (Kullback-Leibler divergence). Suppose that $\mathbb{P}_1$ is absolutely continuous with respect to $\mathbb{P}_2$, the Kullback-Leibler (KL) divergence from $\mathbb{P}_1$ to $\mathbb{P}_2$ is defined as $\mathrm{KL}(\mathbb{P}_1 \parallel \mathbb{P}_2) \triangleq \mathbb{E}_{\mathbb{P}_1}[\log(\mathrm{d}\mathbb{P}_1/\mathrm{d}\mathbb{P}_2)]$, where $\mathrm{d}\mathbb{P}_1/\mathrm{d}\mathbb{P}_2$ is the Radon-Nikodym derivative of $\mathbb{P}_1$ with respect to $\mathbb{P}_2$.

The KL divergence is an ideal choice in our setting for numerous reasons. Previously, DRO problems with a KL ambiguity set often result in tractable finite-dimensional reformulations [5, 19, 6]. More importantly, the manifold of exponential family of distributions equipped with the KL divergence inherits a natural geometry endowed by a dually flat and invariant Riemannian structure [1, Chapter 2]. Furthermore, the KL divergence between two distributions in the same exponential family admits a closed form expression [2, 1].

**Lemma 2.5** (KL divergence between distributions from exponential family). The KL divergence from $\mathbb{Q}_1 \sim f(\,\cdot\,|\theta_1)$ to $\mathbb{Q}_2 \sim f(\,\cdot\,|\theta_2)$ amounts to $\mathrm{KL}(\mathbb{Q}_1 \parallel \mathbb{Q}_2) = \langle\theta_1-\theta_2, \nabla\Psi(\theta_1)\rangle - \Psi(\theta_1) + \Psi(\theta_2)$.

Using the above components, we are now ready to introduce our ambiguity set $\mathbb{B}(\widehat{\mathbb{P}})$ as

$$\mathbb{B}(\widehat{\mathbb{P}}) \triangleq \left\{ \mathbb{Q} \in \mathcal{M}(\mathcal{X}\times\mathcal{Y}) : \begin{array}{l} \exists\mathbb{Q}_X \in \mathcal{M}(\mathcal{X}),\ \exists\theta_c \in \Theta \text{ such that } \mathbb{Q}_{Y|\widehat{x}_c} \sim f(\,\cdot\,|\theta_c) \quad \forall c \\ \mathbb{Q}(\{\widehat{x}_c\}\times A) = \mathbb{Q}_X(\{\widehat{x}_c\})\mathbb{Q}_{Y|\widehat{x}_c}(A) \ \forall c, \forall A \subseteq \mathcal{Y} \text{ measurable} \\ \mathrm{KL}(\mathbb{Q}_{Y|\widehat{x}_c} \parallel \widehat{\mathbb{P}}_{Y|\widehat{x}_c}) \leq \rho_c \quad \forall c \\ \mathrm{KL}(\mathbb{Q}_X \parallel \widehat{\mathbb{P}}_X) + \mathbb{E}_{\mathbb{Q}_X}[\sum_{c=1}^C \rho_c \mathbb{1}_{\widehat{x}_c}(X)] \leq \varepsilon \end{array} \right\} \quad (6)$$

parametrized by a marginal radius $\varepsilon$ and a collection of the conditional radii $\rho_c$. Any distribution $\mathbb{Q} \in \mathbb{B}(\widehat{\mathbb{P}})$ can be decomposed into a marginal distribution $\mathbb{Q}_X$ of the covariate and an ensemble of parametric conditional distributions $\mathbb{Q}_{Y|\widehat{x}_c} \sim f(\,\cdot\,|\theta_c)$ at every event $X = \widehat{x}_c$. The first inequality in (6) restricts the parametric conditional distribution $\mathbb{Q}_{Y|\widehat{x}_c}$ to be in the $\rho_c$-neighborhood from the nominal $\widehat{\mathbb{P}}_{Y|\widehat{x}_c}$ prescribed using the KL divergence, while the second inequality imposes a similar restriction for the marginal distribution $\mathbb{Q}_X$. One can show that for any conditional radii $\rho \in \mathbb{R}_+^C$ satisfying $\sum_{c=1}^C \widehat{p}_c\rho_c \leq \varepsilon$, $\mathbb{B}(\widehat{\mathbb{P}})$ is non-empty with $\widehat{\mathbb{P}} \in \mathbb{B}(\widehat{\mathbb{P}})$. Moreover, if all $\rho$ and $\varepsilon$ are zero, then $\mathbb{B}(\widehat{\mathbb{P}})$ becomes the singleton set $\{\widehat{\mathbb{P}}\}$ that contains only the nominal distribution.

The set $\mathbb{B}(\widehat{\mathbb{P}})$ is a *parametric ambiguity set*: all conditional distributions $\mathbb{Q}_{Y|\widehat{x}_c}$ belong to the same parametric exponential family, and at the same time, the marginal distribution $\mathbb{Q}_X$ is absolutely continuous with respect to a discrete distribution $\widehat{\mathbb{P}}_X$ and hence $\mathbb{Q}_X$ can be parametrized using a $C$-dimensional probability vector.

At first glance, the ambiguity set $\mathbb{B}(\widehat{\mathbb{P}})$ looks intricate and one may wonder whether the complexity of $\mathbb{B}(\widehat{\mathbb{P}})$ is necessary. In fact, it is appealing to consider the ambiguity set

$$\mathcal{B}(\widehat{\mathbb{P}}) \triangleq \left\{ \mathbb{Q} \in \mathcal{M}(\mathcal{X}\times\mathcal{Y}) : \begin{array}{l} \exists\mathbb{Q}_X \in \mathcal{M}(\mathcal{X}),\ \exists\theta_c \in \Theta \text{ such that } \mathbb{Q}_{Y|\widehat{x}_c} \sim f(\,\cdot\,|\theta_c) \quad \forall c \\ \mathbb{Q}(\{\widehat{x}_c\}\times A) = \mathbb{Q}_X(\{\widehat{x}_c\})\mathbb{Q}_{Y|\widehat{x}_c}(A) \ \forall c, \forall A \subseteq \mathcal{Y} \text{ measurable} \\ \mathrm{KL}(\mathbb{Q} \parallel \widehat{\mathbb{P}}) \leq \varepsilon \end{array} \right\} \quad (7)$$

which still preserves the parametric conditional structure and entails only one KL divergence constraint on the *joint* distribution space. Unfortunately, the ambiguity set $\mathcal{B}(\widehat{\mathbb{P}})$ may be overly conservative as pointed out in the following result.

**Proposition 2.6.** Denote momentarily the ambiguity sets (6) and (7) by $\mathbb{B}_{\varepsilon,\rho}(\widehat{\mathbb{P}})$ and $\mathcal{B}_\varepsilon(\widehat{\mathbb{P}})$ to make the dependence on the radii explicit. For any nominal distribution $\widehat{\mathbb{P}}$ of the form (5) and any radius $\varepsilon \in \mathbb{R}_+$, we have

$$\mathcal{B}_\varepsilon(\widehat{\mathbb{P}}) = \bigcup_{\rho\in\mathbb{R}_+^C} \mathbb{B}_{\varepsilon,\rho}(\widehat{\mathbb{P}}).$$

Proposition 2.6 suggests that the ambiguity set $\mathcal{B}(\widehat{\mathbb{P}})$ can be significantly bigger than $\mathbb{B}(\widehat{\mathbb{P}})$, and that the solution of the distributionally robust MLE problem (4) with $\mathbb{B}(\widehat{\mathbb{P}})$ being replaced by $\mathcal{B}(\widehat{\mathbb{P}})$ is potentially too conservative and may lead to undesirable or uninformative results.

The ambiguity set $\mathbb{B}(\widehat{\mathbb{P}})$ requires $1 + C$ parameters, including one marginal radius $\varepsilon$ and $C$ conditional radii $\rho_c$, $c = 1, \ldots, C$, which may be cumbersome to tune in the implementation. Fortunately, by the asymptotic result in Lemma 4.4, the set of radii $\rho_c$ can be tuned simultaneously using the same scaling rate, which will significantly reduce the computational efforts for parameter tuning.

## 3   Tractable Reformulation

We devote this section to study the solution method for the min-max problem (4) by transforming it into a finite dimensional minimization problem. To facilitate the exposition, we denote the ambiguity set for the conditional distribution of $Y$ given $X = \widehat{x}_c$ as

$$\mathbb{B}_{Y|\widehat{x}_c} \triangleq \left\{ \mathbb{Q}_{Y|\widehat{x}_c} \in \mathcal{M}(\mathcal{Y}) : \exists \theta \in \Theta, \ \mathbb{Q}_{Y|\widehat{x}_c}(\cdot) \sim f(\cdot\,|\theta), \ \mathrm{KL}(\mathbb{Q}_{Y|\widehat{x}_c} \,\|\, \widehat{\mathbb{P}}_{Y|\widehat{x}_c}) \leq \rho_c \right\}. \quad (8)$$

As a starting point, we first show the following decomposition of the worst-case expected loss under the ambiguity set $\mathbb{B}(\widehat{\mathbb{P}})$ for any measurable loss function.

**Proposition 3.1** (Worst-case expected loss)**.** Suppose that $\mathbb{B}(\widehat{\mathbb{P}})$ is defined as in (6) for some $\varepsilon \in \mathbb{R}_+$ and $\rho \in \mathbb{R}_+^C$ such that $\sum_{c=1}^C \widehat{p}_c \rho_c \leq \varepsilon$. For any function $L : \mathcal{X} \times \mathcal{Y} \to \mathbb{R}$ measurable, we have

$$\sup_{\mathbb{Q} \in \mathbb{B}(\widehat{\mathbb{P}})} \mathbb{E}_{\mathbb{Q}}\left[L(X, Y)\right] = \left\{ \begin{array}{ll} \inf & \alpha + \beta\varepsilon + \beta \sum_{c=1}^C \widehat{p}_c \exp\left(\beta^{-1}(t_c - \alpha) - \rho_c - 1\right) \\ \mathrm{s.\,t.} & t \in \mathbb{R}^C, \ \alpha \in \mathbb{R}, \ \beta \in \mathbb{R}_{++} \\ & \displaystyle\sup_{\mathbb{Q}_{Y|\widehat{x}_c} \in \mathbb{B}_{Y|\widehat{x}_c}} \mathbb{E}_{\mathbb{Q}_{Y|\widehat{x}_c}}\left[L(\widehat{x}_c, Y)\right] \leq t_c \quad \forall c = 1, \ldots, C. \end{array} \right.$$

Proposition 3.1 leverages the decomposition structure of the ambiguity set $\mathbb{B}(\widehat{\mathbb{P}})$ to reformulate the worst-case expected loss into an infimum problem that involves $C$ constraints, where each constraint is a hypergraph reformulation of a worst-case conditional expected loss under the ambiguity set $\mathbb{B}_{Y|\widehat{x}_c}$. Proposition 3.1 suggests that to reformulate the min-max estimation problem (4), it suffices now to reformulate the worst-case conditional expected log-loss

$$\sup_{\mathbb{Q}_{Y|\widehat{x}_c} \in \mathbb{B}_{Y|\widehat{x}_c}} \mathbb{E}_{\mathbb{Q}_{Y|\widehat{x}_c}}\left[\ell_\lambda(\widehat{x}_c, Y, w)\right] \quad (9)$$

for each value of $\widehat{x}_c$ into a dual infimum problem. Using Lemma 2.5, one can rewrite $\mathbb{B}_{Y|\widehat{x}_c}$ in (8) using the natural parameter representation as

$$\mathbb{B}_{Y|\widehat{x}_c} = \left\{ \mathbb{Q}_{Y|\widehat{x}_c} \in \mathcal{M}(\mathcal{Y}) : \exists \theta \in \Theta, \mathbb{Q}_{Y|\widehat{x}_c}(\cdot) \sim f(\cdot\,|\theta), \left\langle \theta - \widehat{\theta}_c, \nabla\Psi(\theta) \right\rangle - \Psi(\theta) + \Psi(\widehat{\theta}_c) \leq \rho_c \right\}.$$

Since $\Psi$ is convex [2, Lemma 1], it is possible that $\mathbb{B}_{Y|\widehat{x}_c}$ is represented by a non-convex set of natural parameters and hence reformulating (9) is non-trivial. Surprisingly, the next proposition asserts that problem (9) always admits a convex reformulation.

**Proposition 3.2** (Worst-case conditional expected log-loss)**.** For any $\widehat{x}_c \in \mathcal{X}$ and $w \in \mathcal{W}$, the worst-case conditional expected log-loss (9) is equivalent to the univariate convex optimization problem

$$\inf_{\gamma_c \in \mathbb{R}_{++}} \gamma_c\left(\rho_c - \Psi(\widehat{\theta}_c)\right) + \gamma_c\Psi\left(\widehat{\theta}_c - \gamma_c^{-1}\lambda(w, \widehat{x}_c)\right) + \Psi\left(\lambda(w, \widehat{x}_c)\right). \quad (10)$$

A reformulation for the worst-case conditional expected log-loss was proposed in [19]. Nevertheless, the results in [19, Section 5.3] requires that the sufficient statistics $T(y)$ is a linear function of $y$. The reformulation (10), on the other hand, is applicable when $T$ is a *non*linear function of $y$. Examples of exponential family of distributions with nonlinear $T$ are (multivariate) Gaussian, Gamma and Beta distributions. The results from Propositions 3.1 and 3.2 lead to the reformulation of the distributionally robust estimation problem (4), which is the main result of this section.

**Theorem 3.3** (Distributionally robust MLE reformulation)**.** The distributionally robust MLE problem (4) is tantamount to the following finite dimensional optimization problem

$$\begin{array}{ll} \inf & \alpha + \beta\varepsilon + \beta \sum_{c=1}^C \widehat{p}_c \exp(\beta^{-1}(t_c - \alpha) - \rho_c - 1) \\ \mathrm{s.\,t.} & w \in \mathcal{W}, \ \alpha \in \mathbb{R}, \ \beta \in \mathbb{R}_{++}, \ \gamma \in \mathbb{R}_{++}^C, \ t \in \mathbb{R}^C \\ & \gamma_c\left(\rho_c - \Psi(\widehat{\theta}_c)\right) + \gamma_c\Psi\left(\widehat{\theta}_c - \gamma_c^{-1}\lambda(w, \widehat{x}_c)\right) + \Psi\left(\lambda(w, \widehat{x}_c)\right) \leq t_c \quad \forall c = 1, \ldots, C. \end{array} \quad (11)$$

In generalized linear models with $\lambda : (w, x) \mapsto w^\top x$ and $\mathcal{W}$ being convex, problem (11) is convex. Below we show how the Poisson and logistic regression models fit within this framework.

**Example 3.4** (Poisson counting model)**.** The Poisson counting model with the ground measure $\nu$ being a counting measure on $\mathcal{Y} = \mathbb{N}$, the sufficient statistic $T(y) = y$, the natural parameter space $\Theta = \mathbb{R}$ and the log-partition function $\Psi(\theta) = \exp(\theta)$. If $\lambda(w, x) = w^\top x$, we have

$$Y|X = x \sim \mathrm{Poisson}\big(w_0^\top x\big), \qquad \mathbb{P}(Y = k | X = x) = (k!)^{-1} \exp(k w_0^\top x - e^{w_0^\top x}).$$

The distributionally robust MLE is equivalent to the following convex optimization problem

$$
\begin{aligned}
\inf \quad & \alpha + \beta \varepsilon + \beta \textstyle\sum_{c=1}^{C} \widehat{p}_c \exp\big(\beta^{-1}(t_c - \alpha) - \rho_c - 1\big) \\
\mathrm{s.\,t.} \quad & w \in \mathcal{W}, \ \alpha \in \mathbb{R}, \ \beta \in \mathbb{R}_{++}, \ \gamma \in \mathbb{R}_{++}^{C}, \ t \in \mathbb{R}^{C} \\
& \gamma_c\big(\rho_c - \exp(\widehat{\theta}_c)\big) + \gamma_c \exp\big(\widehat{\theta}_c - w^\top \widehat{x}_c / \gamma_c\big) + \exp\big(w^\top \widehat{x}_c\big) \le t_c \quad \forall c = 1, \ldots, C.
\end{aligned}
\tag{12}
$$

**Example 3.5** (Logistic regression)**.** The logistic regression model is specified with $\nu$ being a counting measure on $\mathcal{Y} = \{0, 1\}$, the sufficient statistic $T(y) = y$, the natural parameter space $\Theta = \mathbb{R}$ and the log-partition function $\Psi(\theta) = \log\big(1 + \exp(\theta)\big)$. If $\lambda(w, x) = w^\top x$, we have

$$Y|X = x \sim \mathrm{Bernoulli}\big((1 + \exp(-w_0^\top x))^{-1}\big), \qquad \mathbb{P}(Y = 1 | X = x) = (1 + \exp(-w_0^\top x))^{-1}.$$

The distributionally robust MLE is equivalent to the following convex optimization problem

$$
\begin{aligned}
\inf \quad & \alpha + \beta \varepsilon + \beta \textstyle\sum_{c=1}^{C} \widehat{p}_c \exp\big(\beta^{-1}(t_c - \alpha) - \rho_c - 1\big) \\
\mathrm{s.\,t.} \quad & w \in \mathcal{W}, \ \alpha \in \mathbb{R}, \ \beta \in \mathbb{R}_{++}, \ \gamma \in \mathbb{R}_{++}^{C}, \ t \in \mathbb{R}^{C} \\
& \gamma_c\big(\rho_c - \log(1 + \exp(\widehat{\theta}_c))\big) + \gamma_c \log\big(1 + \exp(\widehat{\theta}_c - w^\top \widehat{x}_c / \gamma_c)\big) + \log\big(1 + \exp(w^\top \widehat{x}_c)\big) \le t_c \ \forall c.
\end{aligned}
\tag{13}
$$

Problems (12) and (13) can be solved by exponential conic solvers such as ECOS [12] and MOSEK [24].

## 4 Theoretical Analysis

In this section, we provide an in-depth theoretical analysis of our estimator. We first show that our proposed estimator is tightly connected to several existing regularization schemes.

**Proposition 4.1** (Connection to the adversarial reweighting scheme)**.** Suppose that $\widehat{x}_i$ are distinct and $\rho_c = 0$ for any $c = 1, \ldots, N$. If $\widehat{\mathbb{P}}$ is of the form (5) and chosen according to Example 2.2, then the distributionally robust estimation problem (4) is equivalent to

$$\min_{w \in \mathcal{W}} \ \sup_{\mathbb{Q} : \mathrm{KL}(\mathbb{Q} \| \widehat{\mathbb{P}}^{\mathrm{emp}}) \le \varepsilon} \ \mathbb{E}_{\mathbb{Q}}[\ell_\lambda(X, Y, w)].$$

Proposition 4.1 asserts that by setting the conditional radii to zero, we can recover the robust estimation problem where the ambiguity set is a KL ball around the empirical distribution $\widehat{\mathbb{P}}^{\mathrm{emp}}$, which has been shown to produce the adversarial reweighting effects [23, 6]. Recently, it has been shown that distributionally robust optimization using $f$-divergences is statistically related to the variance regularization of the empirical risk minimization problem [26]. Our proposed estimator also admits a variance regularization surrogate, as asserted by the following proposition.

**Proposition 4.2** (Variance regularization surrogate)**.** Suppose that $\Psi$ has locally Lipschitz continuous gradients. For any fixed $\widehat{\theta}_c \in \Theta$, $c = 1, \ldots, C$, there exists a constant $m > 0$ that depends only on $\Psi$ and $\widehat{\theta}_c$, $c = 1, \ldots, C$, such that for any $w \in \mathcal{W}$ and $\varepsilon \ge \sum_{c=1}^{C} \widehat{p}_c \rho_c$, we have

$$\sup_{\mathbb{Q} \in \mathbb{B}(\widehat{\mathbb{P}})} \ \mathbb{E}_{\mathbb{Q}}[\ell_\lambda(X, Y, w)] \le \mathbb{E}_{\widehat{\mathbb{P}}}[\ell_\lambda(X, Y, w)] + \kappa_1 \sqrt{\mathrm{Var}_{\widehat{\mathbb{P}}}\big(\ell_\lambda(X, Y, w)\big)} + \kappa_2 \|\lambda(w, \widehat{x}_c)\|_2,$$

where $\kappa_1 = \sqrt{2\varepsilon}/(\min_c \sqrt{\widehat{p}_c})$ and $\kappa_2 = \sqrt{2 \max_c \rho_c / m}$.

One can further show that for sufficiently small $\rho_c$, the value of $m$ is proportional to the inverse of the local Lipschitz constant of $\nabla \Psi$ at $\widehat{\theta}_c$, in which case $\kappa_2$ admits an explicit expression (see Appendix D). Next, we show that our robust estimator is also consistent, which is a highly desirable statistical property.

**Theorem 4.3** (Consistency). Assume that $w_0$ is the unique solution of the problem $\min_{w \in \mathcal{W}} \mathbb{E}_{\mathbb{P}}[\ell_\lambda(X, Y, w)]$, where $\mathbb{P}$ denotes the true distribution. Assume that $\mathcal{X}$ has finite cardinality, $\Theta = \mathbb{R}^p$, $\Psi$ has locally Lipschitz continuous gradients, and $\ell_\lambda(x, y, w)$ is convex in $w$ for each $x$ and $y$. If $\widehat{\theta}_c \xrightarrow{p.} \lambda(w_0, \widehat{x}_c)$ for each $c$, $\varepsilon \to 0$, $\rho_c \to 0$ and $\varepsilon \geq \sum_{c=1}^{C} \widehat{p}_c \rho_c$ with probability going to 1, then the distributionally robust estimator $w^\star$ that solves (4) exists with probability going to 1, and $w^\star \xrightarrow{p.} w_0$.

One can verify that choosing $\widehat{\theta}_c$ using Examples 2.2 and 2.3 will satisfy the condition $\widehat{\theta}_c \xrightarrow{p.} \lambda(w_0, \widehat{x}_c)$, and as a direct consequence, choosing $\widehat{\mathbb{P}}$ following these two examples will result in a consistent estimator under the conditions of Theorem 4.3.

We now consider the asymptotic scaling rate of $\rho_c$ as the number $N_c$ of samples with the same covariate $\widehat{x}_c$ tends to infinity. Lemma 4.4 below asserts that $\rho_c$ should scale at the rate $N_c^{-1}$. Based on this result, we can set $\rho_c = a N_c^{-1}$ for all $c$, where $a > 0$ is a tuning parameter. This reduces significantly the burden of tuning $\rho_c$ down to tuning a single parameter $a$.

**Lemma 4.4** (Joint asymptotic convergence). Suppose that $|\mathcal{X}| = C$ with $\mathbb{P}(X = \widehat{x}_c) > 0$. Let $\theta_c = \lambda(w_0, \widehat{x}_c)$ and $\widehat{\mathbb{P}}$ be defined as in Example 2.2. Let $V_c = D_c \mathrm{Cov}_{f(\cdot | \theta_c)}(T(Y)) D_c^\top$, where $D_c = J(\nabla \Psi)^{-1} (\mathbb{E}_{f(\cdot | \theta_c)}[T(Y)])$ and $J$ denotes the Jacobian operator. Then the following joint convergence holds

$$\left(N_1 \times \mathrm{KL}(f(\cdot | \theta_1) \| f(\cdot | \widehat{\theta}_1)), \ldots, N_C \times \mathrm{KL}(f(\cdot | \theta_C) \| f(\cdot | \widehat{\theta}_C))\right)^\top \xrightarrow{d.} Z \qquad \text{as} \qquad N \to \infty, \tag{14}$$

where $Z = (Z_1, \ldots, Z_C)^\top$ with $Z_c = \frac{1}{2} R_c^\top \nabla^2 \Psi(\theta_c) R_c$, $R_c$ are independent and $R_c \sim \mathcal{N}(0, V_c)$.

Assuming $w_{MLE}$ that solves (3a) is asymptotically normal with square-root convergence rate, we remark that the asymptotic joint convergence (14) also holds for $\widehat{\mathbb{P}}$ in Example 2.3, though in this case the limiting distribution $Z$ takes a more complex form that can be obtained by the delta method.

Finally, we study the structure of the worst-case distribution $\mathbb{Q}^\star = \arg\max_{\mathbb{Q} \in \mathbb{B}(\widehat{\mathbb{P}})} \mathbb{E}_{\mathbb{Q}}[\ell_\lambda(X, Y, w)]$ for any value of input $w$. This result explicitly quantifies how the adversary will generate the adversarial distribution adapted to any estimate $w$ provided by the statistician.

**Theorem 4.5** (Worst-case joint distribution). Given $\rho \in \mathbb{R}_+^C$ and $\varepsilon \in \mathbb{R}_+$ such that $\sum_{c=1}^C \widehat{p}_c \rho_c \leq \varepsilon$. For any $w$ and $c = 1, \ldots, C$, let $\mathbb{Q}_{Y|\widehat{x}_c}^\star \sim f(\cdot | \theta_c^\star)$ with $\theta_c^\star = \widehat{\theta}_c - \lambda(w, \widehat{x}_c)/\gamma_c^\star$, where $\gamma_c^\star > 0$ is the solution of the nonlinear equation

$$\Psi\left(\widehat{\theta}_c - \gamma^{-1}\lambda(w, \widehat{x}_c)\right) + \gamma^{-1}\left\langle \nabla\Psi\left(\widehat{\theta}_c - \gamma^{-1}\lambda(w, \widehat{x}_c)\right), \lambda(w, \widehat{x}_c)\right\rangle = \Psi(\widehat{\theta}_c) - \rho_c,$$

and let $t_c^\star = \Psi(\lambda(w, \widehat{x}_c)) - \left\langle \nabla\Psi(\theta_c^\star), \lambda(w, \widehat{x}_c)\right\rangle$. Let $\alpha^\star \in \mathbb{R}$ and $\beta^\star \in \mathbb{R}_{++}$ be the solution of the following system of nonlinear equations

$$\sum_{c=1}^C \widehat{p}_c \exp\left(\beta^{-1}(t_c^\star - \alpha) - \rho_c - 1\right) - 1 = 0$$
$$\sum_{c=1}^C \widehat{p}_c (t_c^\star - \alpha) \exp\left(\beta^{-1}(t_c^\star - \alpha) - \rho_c - 1\right) - (\varepsilon + 1)\beta = 0,$$

then the worst-case distribution is $\mathbb{Q}^\star = \sum_{c=1}^C \widehat{p}_c \exp\left((\beta^\star)^{-1}(t_c^\star - \alpha^\star) - \rho_c - 1\right) \delta_{\widehat{x}_c} \otimes \mathbb{Q}_{Y|\widehat{x}_c}^\star$.

Notice that $\mathbb{Q}^\star$ is decomposed into a worst-case marginal distribution of $X$ supported on $\widehat{x}_c$ and a collection of worst-case conditional distributions $\mathbb{Q}_{Y|\widehat{x}_c}^\star$.

## 5 Numerical Experiments

We now showcase the abilities of the proposed framework in the distributionally robust Poisson and logistic regression settings using a combination of simulated and empirical experiments. All optimization problems are modeled in MATLAB using CVX [16] and solved by the exponential conic solver MOSEK [24] on an Intel i7 CPU (1.90GHz) computer. Optimization problems (12) and (13) are solved in under 3 seconds for all instances both in the simulated and empirical experiments. The MATLAB code is available at https://github.com/angelosgeorghiou/DR-Parametric-MLE.

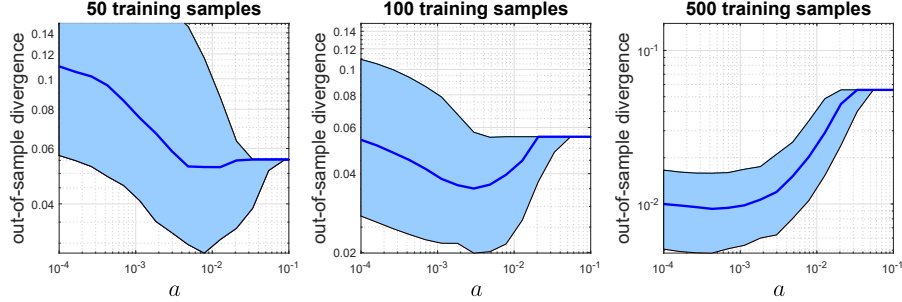

Figure 1: Median (solid blue line) and the 10th-90th percentile region (shaded) of out-of-sample divergence loss collected from 100 independent runs.

| | | $N = 50$ | $N = 100$ | $N = 500$ |
|---|---|---|---|---|
| CI$_{95\%}$ | 100(DRO-MLE)/MLE | $-69.84 \pm 3.33\%$ | $-52.66 \pm 3.98\%$ | $-22.25 \pm 4.11\%$ |
| | 100(DRO $- L_1$)/$L_1$ | $-22.59 \pm 4.02\%$ | $-19.90 \pm 4.16\%$ | $-13.20 \pm 3.38\%$ |
| | 100(DRO $- L_2$)/$L_2$ | $-14.98 \pm 4.08\%$ | $-9.98 \pm 4.14\%$ | $-5.62 \pm 2.83\%$ |
| CVaR$_{5\%}$ | MLE | 0.4906 | 0.1651 | 0.0246 |
| | $L_1$ | 0.0967 | 0.0742 | 0.0195 |
| | $L_2$ | 0.0894 | 0.0692 | 0.0176 |
| | DRO | 0.0547 | 0.0518 | 0.0172 |

Table 1: Comparison between the DRO estimator with the other methods. Lower values are better.

## 5.1 Poisson Regression

We will use simulated experiments to demonstrate the behavior of the tuning parameters and to compare the performance of our estimator with regard to other established methods. We assume that the true distribution $\mathbb{P}$ is discrete, the 10-dimensional covariate $X$ is supported on $K = 100$ points and their locations $\widehat{x}_k$ are generated i.i.d. using a standard normal distribution. We then generate a $K$-dimensional vector whose components are i.i.d. uniform over $M_k \in [0, 10000]$, then normalize it to get the probability vector $p_k = M_k/M$ of the true marginal distribution of $X$. The value $w_0$ that determines the true conditional distribution $\mathbb{P}_{Y|X}$ via the generative model (1) is assigned to $w_0 = \tilde{w}/\|\tilde{w}\|_1$, where $\tilde{w}$ is drawn randomly from a 10-dimensional standard normal distribution.

Our experiment comprises 100 simulation runs. In each run we generate $N \in \{50, 100, 500\}$ training samples i.i.d. from $\mathbb{P}$ and use the MLE-compatible nominal distribution $\widehat{\mathbb{P}}$ of the form (5) as in Example 2.3. We calibrate the regression model (12) by tuning $\rho_c = aN_c^{-1}$ with $a \in [10^{-4}, 1]$ and $\varepsilon \in [\sum_{c=1}^C \widehat{p}_c \rho_c, 1]$, both using a logarithmic scale with 20 discrete points. The quality of an estimate $w^\star$ with respect to the true distribution $\mathbb{P}$ is evaluated by the out-of-sample divergence loss

$$\mathbb{E}_{\mathbb{P}_X}[\mathrm{KL}(\mathbb{P}_{Y|X} \| \mathbb{Q}_{w^\star, Y|X})] = \sum_{k=1}^K p_k \big( \exp(w_0^\top \widehat{x}_k)\big((w_0 - w^\star)^\top \widehat{x}_k - 1\big) + \exp(\widehat{x}_k^\top w^\star)\big).$$

In the first numerical experiment, we fix the marginal radius $\varepsilon = 1$ and examine how tuning the conditional radii $\rho_c$ can improve the quality of the estimator. Figure 1 shows the 10th, 50th and 90th percentile of the out-of-sample divergence for different samples sizes. If the constant $a$ is chosen judiciously, incorporating the uncertainty in the conditional distribution can reduce the out-of-sample divergence loss by 17.65%, 10.55% and 1.82% for $N = 50, 100$ and 500, respectively.

Next, we compare the performance of our proposed estimator to the $w_{MLE}$ that solves (3a) and the 1-norm ($L_1$) and 2-norm ($L_2$) MLE regularization, where the regularization weight takes values in $[10^{-4}, 1]$ on the logarithmic scale with 20 discrete points. In each run, we choose the optimal parameters that give the lowest of out-of-sample divergence for each method, and construct the empirical distribution of the out-of-sample divergence collected from 100 runs. Table 1 reports the 95% confidence intervals of $100(\mathrm{DRO}-\mathrm{MLE})/\mathrm{MLE}$, $100(\mathrm{DRO}-L_1)/L_1$ and $100(\mathrm{DRO}-L_2)/L_2$, as well as the 5% Conditional Value-at-Risk (CVaR). Our approach delivers lower out-of-sample divergence loss compared to the other methods, and additionally ensures a lower value of CVaR for all sample sizes. This improvement is particularly evident in small sample sizes.

| | AUC | | | | | CCR | | | | |
|---|---|---|---|---|---|---|---|---|---|---|
| Dataset | DRO | KL | $L_1$ | $L_2$ | MLE | DRO | KL | $L_1$ | $L_2$ | MLE |
| australian ($N = 690$, $n = 14$) | **92.74** | 92.62 | 92.73 | 92.71 | 92.61 | **85.75** | 85.72 | 85.52 | 85.60 | 85.72 |
| banknote ($N = 1372$, $n = 4$) | **98.46** | **98.46** | 98.43 | 98.45 | 98.45 | 94.31 | 94.32 | 94.16 | **94.35** | 94.32 |
| climate ($N = 540$, $n = 18$) | 94.30 | 82.77 | **94.85** | 94.13 | 82.76 | **95.04** | 93.89 | 94.85 | 94.83 | 93.89 |
| german ($N = 1000, n = 19$) | **75.75** | 75.68 | 75.74 | 75.74 | 75.67 | 73.86 | **74.05** | 73.82 | 73.70 | **74.05** |
| haberman ($N = 306, n = 3$) | 66.86 | 67.21 | **69.19** | 68.17 | 67.20 | **73.83** | 73.80 | 73.20 | 73.18 | 73.80 |
| housing ($N = 506, n = 13$) | **76.24** | 75.73 | 75.37 | 75.57 | 75.73 | 91.65 | 91.70 | **92.68** | 92.65 | 91.70 |
| ILPD ($N = 583$, $n = 10$) | **74.01** | 73.66 | 73.56 | 73.77 | 73.66 | 71.11 | 71.07 | 71.68 | **71.79** | 71.07 |
| mammo. ($N = 830 \, n = 5$) | **87.73** | 87.72 | 87.70 | 87.68 | 87.71 | 81.00 | **81.20** | 80.99 | 80.94 | **81.20** |

Table 2: Average area under the curve (AUC) and correct classification rates (CCR) on UCI datasets ($m = 1$).

## 5.2 Logistic Regression

We now study the performance of our proposed estimation in a classification setting using data sets from the UCI repository [13]. We compare four different models: our proposed DRO estimator (13), the $w_{MLE}$ that solves (3a), the 1-norm ($L_1$) and 2-norm ($L_2$) MLE regularization. In each independent trial, we randomly split the data into train-validation-test set with proportion 50%-25%-25%. For our estimator, we calibrate the regression model (13) by tuning $\rho_c = aN_c^{-1}$ with $a \in [10^{-4}, 10]$ using a logarithmic scale with 10 discrete points and setting $\varepsilon = 2\sum_{c=1}^{C} \widehat{p}_c \rho_c$. Similarly, for the $L_1$ and $L_2$ regularization, we calibrate the regularization weight from $[10^{-4}, 1]$ on the logarithmic scale with 10 discrete points. Leveraging Proposition 4.1, we also compare our approach versus the DRO *non*parametric Kullback-Leibler (KL) MLE by setting $\rho_c = 0$ and tune only with $\varepsilon \in [10^{-4}, 10]$ with 10 logarithmic scale points. The performance of the methods was evaluated on the testing data using two popular metrics: the correct classification rate (CCR) with a threshold level of 0.5, and the area under the receiver operating characteristics curve (AUC). Table 2 reports the performance of each method averaged over 100 runs. One can observe that our estimator performs reasonably well compared to other regularization techniques in both performance metrics.

**Remark 5.1** (Uncertainty in $\widehat{x}_c$)**.** The absolute continuity condition of the KL divergence implies that our proposed model cannot hedge against the error in the covariate $\widehat{x}_c$. It is natural to ask which model can effectively cover this covariate error. Unfortunately, answering this question needs to overcome two technical difficulties: first, the log-partition function $\Psi$ is convex; second, there are multiplicative terms between $X$ and $Y$ in the objective function. Maximizing over the $X$ space to find the worst-case covariate is thus difficult. Alternatively, one can think of perturbing each $\widehat{x}_c$ in a finite set but this approach will lead to trivial modifications of the constraints of problem (11).

## Broader Impact

This is a theoretical contribution which relates to arguably the single most popular class of statistical estimators, namely, maximum likelihood estimators. We provide a novel view of these types of estimators, by introducing robustness in their design. This robustness layer enables the use of these types of estimators in the context of (adversarially) contaminated data, which has been a longstanding issue in research. We believe that this paper makes an important contribution to multiple areas, including but not limited to: adversarial machine learning, convex optimization, distributionally robust optimization, and game theory. The fact that our proposed estimator can be computed efficiently by state-of-the-art solvers sheds light on its wide applicability in general academia and industry setting.

For example, our robust maximum likelihood estimators could potentially be used in situations in which data sets of different types of domains are combined to estimate key performance indicators for decision makers (e.g. collecting data from different types of demand functions in a business setting, or different social impact measures in a public policy setting). This enables the decision maker to design strategies that are robust against changes in business or public circumstances, thereby creating a positive social impact.

In addition, with regard to human resource development, this work will be integrated as a part of the tools that we intend to teach in Ph.D. courses, thus positively impacting the training of the workforce in academia and general industries. Questions remaining for issues like ethical implications of our

estimator, which we intend to explore in the future under the framework such as differential privacy and algorithmic fairness.

**Acknowledgments.** Material in this paper is based upon work supported by the Air Force Office of Scientific Research under award number FA9550-20-1-0397. Additional support is gratefully acknowledged from NSF grants 1915967, 1820942, 1838676 and from the China Merchant Bank.

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
