[Supplementary Material]

# Appendix
# Distributionally Robust Parametric Maximum Likelihood Estimation

**Viet Anh Nguyen**      **Xuhui Zhang**      **José Blanchet**
Stanford University, United States
{viet-anh.nguyen, xuhui.zhang, jose.blanchet}@stanford.edu

**Angelos Georghiou**
University of Cyprus, Cyprus
georghiou.angelos@ucy.ac.cy

This appendix is organized as follows. Section A-C provide the detailed proofs for all the technical results in the main paper. Section D provides further discussion on the variance regularization surrogate result in Proposition 4.2.

## A    Proofs of Section 2

*Proof of Example 2.2.* We note that

$$\min_{w \in \mathcal{W}} \; \mathbb{E}_{\widehat{\mathbb{P}}}[\ell_\lambda(X,Y,w)] = \min_{w \in \mathcal{W}} \sum_{c=1}^{C} \widehat{p}_c \left( \Psi(\lambda(w,\widehat{x}_c)) - \left\langle \mathbb{E}_{\widehat{\mathbb{P}}_{Y|\widehat{x}_c}}[T(Y)], \lambda(w,\widehat{x}_c) \right\rangle \right)$$

$$= \min_{w \in \mathcal{W}} \sum_{c=1}^{C} \widehat{p}_c \left( \Psi(\lambda(w,\widehat{x}_c)) - \left\langle \nabla\Psi(\widehat{\theta}_c), \lambda(w,\widehat{x}_c) \right\rangle \right).$$

If $\widehat{\theta}_c = (\nabla\Psi)^{-1}\big((N_c)^{-1} \sum_{\widehat{x}_i = \widehat{x}_c} T(\widehat{y}_i)\big)$, then we have

$$\min_{w \in \mathcal{W}} \; \mathbb{E}_{\widehat{\mathbb{P}}}[\ell_\lambda(X,Y,w)] = \min_{w \in \mathcal{W}} \sum_{c=1}^{C} \widehat{p}_c \left( \Psi(\lambda(w,\widehat{x}_c)) - \left\langle \frac{\sum_{\widehat{x}_i = \widehat{x}_c} T(\widehat{y}_i)}{N_c}, \lambda(w,\widehat{x}_c) \right\rangle \right)$$

$$= \min_{w \in \mathcal{W}} \frac{1}{N} \sum_{i=1}^{N} \left( \Psi(\lambda(w,\widehat{x}_i)) - \left\langle T(\widehat{y}_i), \lambda(w,\widehat{x}_i) \right\rangle \right),$$

where we used $\widehat{p}_c = N_c/N$. Therefore $w_{MLE}$ solves $\min_{w \in \mathcal{W}} \; \mathbb{E}_{\widehat{\mathbb{P}}}[\ell_\lambda(X,Y,w)]$.    $\square$

*Proof of Example 2.3.* We find

$$\min_{w \in \mathcal{W}} \; \mathbb{E}_{\widehat{\mathbb{P}}}[\ell_\lambda(X,Y,w)] = \min_{w \in \mathcal{W}} \sum_{c=1}^{C} \widehat{p}_c \left( \Psi(\lambda(w,\widehat{x}_c)) - \left\langle \mathbb{E}_{\widehat{\mathbb{P}}_{Y|\widehat{x}_c}}[T(Y)], \lambda(w,\widehat{x}_c) \right\rangle \right)$$

$$\geq \sum_{c=1}^{C} \widehat{p}_c \min_{w_c \in \mathcal{W}} \left( \Psi(\lambda(w_c,\widehat{x}_c)) - \left\langle \mathbb{E}_{\widehat{\mathbb{P}}_{Y|\widehat{x}_c}}[T(Y)], \lambda(w_c,\widehat{x}_c) \right\rangle \right)$$

$$= \sum_{c=1}^{C} \widehat{p}_c \left( \Psi(\lambda(w_{MLE},\widehat{x}_c)) - \left\langle \mathbb{E}_{\widehat{\mathbb{P}}_{Y|\widehat{x}_c}}[T(Y)], \lambda(w_{MLE},\widehat{x}_c) \right\rangle \right),$$

where the first equality follows from the definition of the log-loss function $\ell_\lambda$, the inequality follows because $\widehat{p}_c > 0$, and the last equality follows because of the convex conjugate relationship that implies the optimal solution $w_c^\star$ should satisfy

$$\nabla\Psi\big(\lambda(w_c^\star, \widehat{x}_c)\big) = \mathbb{E}_{\widehat{\mathbb{P}}_{Y|\widehat{x}_c}}[T(Y)] = \nabla\Psi\big(\lambda(w_{MLE}, \widehat{x}_c)\big) \implies w_c^\star = w_{MLE}.$$

This implies that $w_{MLE}$ solves $\min_{w\in\mathcal{W}} \ \mathbb{E}_{\widehat{\mathbb{P}}}[\ell_\lambda(X, Y, w)]$ and completes the proof. $\qquad\square$

*Proof of Proposition 2.6.* Fix any set of conditional radii $\rho \in \mathbb{R}_+^C$. If $\mathbb{B}_{\varepsilon,\rho}(\widehat{\mathbb{P}})$ is empty then it is trivial that $\mathbb{B}_{\varepsilon,\rho}(\widehat{\mathbb{P}}) \subset \mathcal{B}_\varepsilon(\widehat{\mathbb{P}})$. Suppose that $\mathbb{B}_{\varepsilon,\rho}(\widehat{\mathbb{P}})$ is non-empty and pick any $\mathbb{Q} \in \mathbb{B}_{\varepsilon,\rho}(\widehat{\mathbb{P}})$. By definition of the set $\mathbb{B}_{\varepsilon,\rho}(\widehat{\mathbb{P}})$, $\mathbb{Q}$ can be decomposed into a marginal $\mathbb{Q}_X$ and a collection of conditional measures $\mathbb{Q}_{Y|\widehat{x}_c}$. Furthermore, because $\varepsilon$ is finite, the marginal $\mathbb{Q}_X$ should be absolutely continuous with respect to $\widehat{\mathbb{P}}_X$. We have

$$\mathrm{KL}(\mathbb{Q} \parallel \widehat{\mathbb{P}}) = \mathrm{KL}(\mathbb{Q}_X \parallel \widehat{\mathbb{P}}_X) + \mathbb{E}_{\mathbb{Q}_X}[\mathrm{KL}(\mathbb{Q}_{Y|X} \parallel \widehat{\mathbb{P}}_{Y|X})]$$

$$\leq \mathrm{KL}(\mathbb{Q}_X \parallel \widehat{\mathbb{P}}_X) + \mathbb{E}_{\mathbb{Q}_X}[\sum_{c=1}^{C} \rho_c \mathbb{1}_{\widehat{x}_c}(X)] \leq \varepsilon,$$

where the equality is from the chain rule of the conditional relative entropy [10, Lemma 7.9]. The first inequality follows from the fact that $\mathrm{KL}(\mathbb{Q}_{Y|\widehat{x}_c} \parallel \widehat{\mathbb{P}}_{Y|\widehat{x}_c}) \leq \rho_c$ for every $c$. The second inequality follows from the last constraint defining the set $\mathbb{B}_{\varepsilon,\rho}(\widehat{\mathbb{P}})$. This implies that $\mathbb{Q} \in \mathcal{B}_\varepsilon(\widehat{\mathbb{P}})$, and because $\mathbb{Q}$ was chosen arbitrarily, we have $\mathbb{B}_{\varepsilon,\rho} \subseteq \mathcal{B}_\varepsilon(\widehat{\mathbb{P}})$. As a consequence, $\bigcup_{\rho\in\mathbb{R}_+^C} \mathbb{B}_{\varepsilon,\rho}(\widehat{\mathbb{P}}) \subseteq \mathcal{B}_\varepsilon(\widehat{\mathbb{P}})$.

Regarding the reverse relation, pick an arbitrary $\mathbb{Q} \in \mathcal{B}_\varepsilon(\widehat{\mathbb{P}})$ which admits the decomposition into a marginal $\mathbb{Q}_X$ and conditional measures $\mathbb{Q}_{Y|\widehat{x}_c}$. By setting the conditional radii $\rho \in \mathbb{R}_+^C$ with $\rho_c = \mathrm{KL}(\mathbb{Q}_{Y|\widehat{x}_c} \parallel \widehat{\mathbb{P}}_{Y|\widehat{x}_c})$ for every $c$, one can verify using the chain rule of the conditional relative entropy that $\mathbb{Q} \in \mathbb{B}_{\varepsilon,\rho}(\widehat{\mathbb{P}})$. This implies that $\mathbb{B}_{\varepsilon,\rho}(\widehat{\mathbb{P}}) \subseteq \bigcup_{\rho\in\mathbb{R}_+^C} \mathbb{B}_{\varepsilon,\rho}(\widehat{\mathbb{P}})$.

Concerning the last statement, notice that the condition $\sum_{c=1}^{C} \widehat{p}_c \rho_c \leq \varepsilon$ implies that $\widehat{\mathbb{P}} \in \mathbb{B}_{\varepsilon,\rho}(\widehat{\mathbb{P}})$ and thus $\mathbb{B}_{\varepsilon,\rho}(\widehat{\mathbb{P}})$ is non-empty. The proof is complete. $\qquad\square$

# B   Proofs of Section 3

The proof of Proposition 3.1 relies on the following preliminary result.

**Lemma B.1.** Let $\widehat{p} \in \mathbb{R}_{++}^C$ be a probability vector summing up to one. For any $\varepsilon \in \mathbb{R}_+$ and $\rho \in \mathbb{R}_+^C$ satisfying $\sum_{c=1}^{C} \widehat{p}_c \rho_c \leq \varepsilon$, the finite dimensional set

$$\mathcal{Q} \triangleq \left\{ q \in \mathbb{R}_+^C : \sum_{c=1}^{C} q_c = 1, \ \sum_{c=1}^{C} q_c(\log q_c - \log\widehat{p}_c + \rho_c) \leq \varepsilon \right\} \qquad (A.1)$$

is compact and convex. Moreover, the support function $h_\mathcal{Q}$ of $\mathcal{Q}$ satisfies

$$\forall t \in \mathbb{R}^C : \quad h_\mathcal{Q}(t) \triangleq \sup_{q\in\mathcal{Q}} q^\top t = \inf_{\alpha\in\mathbb{R}, \ \beta\in\mathbb{R}_{++}} \left\{ \alpha + \beta\varepsilon + \beta \sum_{c=1}^{C} \widehat{p}_c \exp\left( \frac{t_c - \alpha}{\beta} - \rho_c - 1 \right) \right\}.$$

*Proof of Lemma B.1.* The function $\mathbb{R}_+^C \ni q \mapsto \sum_{c=1}^{C} q_c(\log q_c - \log\widehat{p}_c + \rho_c) \in \mathbb{R}_+$ is continuous and convex, hence, the set $\{q \in \mathbb{R}_+^C : \sum_{c=1}^{C} q_c(\log q_c - \log\widehat{p}_c + \rho_c) \leq \varepsilon\}$ is closed and convex. Consequentially, $\mathcal{Q}$ can be written as the intersection between a simplex (thus compact and convex) and a closed, convex set, so $\mathcal{Q}$ is compact and convex.

The proof of the support function of $\mathcal{Q}$ proceeds in 2 steps. First, we prove the support function for the $\epsilon$-inflated set

$$\mathcal{Q}_\epsilon = \left\{ q \in \mathbb{R}_+^C : \sum_{c=1}^{C} q_c = 1, \ \sum_{c=1}^{C} q_c(\log q_c - \log\widehat{p}_c + \rho_c) \leq \varepsilon + \epsilon \right\}$$

with the right-hand side of the last constraint being inflated with $\epsilon \in \mathbb{R}_{++}$. In the second step, we use a limit argument to show that the support function of $\mathcal{Q}$ is attained as the limit of the support function of $\mathcal{Q}_\epsilon$ as $\epsilon$ tends to 0.

Reminding that $\Delta$ is the $C$-dimensional simplex. For any $t \in \mathbb{R}^C$ and any $\epsilon \in \mathbb{R}_{++}$, by the definition of the support function, we have for every $t \in \mathbb{R}^C$

$$
h_{\mathcal{Q}_\epsilon}(t) =
\begin{cases}
\sup & q^\top t \\
\text{s.t.} & q \in \Delta, \ \sum_{c=1}^{C} q_c(\log q_c - \log \widehat{p}_c + \rho_c) \leq \varepsilon + \epsilon
\end{cases}
\tag{A.2a}
$$

$$
= \sup_{q \in \Delta} \ \inf_{\beta \in \mathbb{R}_+} \ q^\top t + \beta\Big(\varepsilon + \epsilon - \sum_{c=1}^{C} q_c\big(\log q_c - \log \widehat{p}_c + \rho_c\big)\Big)
$$

$$
= \inf_{\beta \in \mathbb{R}_+} \ \sup_{q \in \Delta} \ q^\top t + \beta\Big(\varepsilon + \epsilon - \sum_{c=1}^{C} q_c\big(\log q_c - \log \widehat{p}_c + \rho_c\big)\Big),
\tag{A.2b}
$$

where the interchange of the sup-inf operators in (A.2b) is justified by strong duality [6, Proposition 5.3.1] because $\widehat{p}$ constitutes a Slater point of the set $\mathcal{Q}_\epsilon$. By Berge's maximum theorem [5], the optimal value of the inner supremum problem is a continuous function in $\beta$ because the simplex $\Delta$ is compact and the objective function is continuous in the decision variable $q$. As a consequence, we can restrict $\beta \in \mathbb{R}_{++}$ without any loss of optimality. Because $\Delta$ is prescribed using linear constraints, strong duality implies that

$$
h_{\mathcal{Q}_\epsilon}(t) = \inf_{\alpha \in \mathbb{R}, \ \beta \in \mathbb{R}_{++}} \left\{ \alpha + \beta(\varepsilon + \epsilon) + \sup_{q \in \mathbb{R}_+^C} \sum_{c=1}^{C} q_c(t_c - \alpha + \beta \log \widehat{p}_c - \beta \rho_c - \beta \log q_c) \right\}
$$

$$
= \inf_{\alpha \in \mathbb{R}, \ \beta \in \mathbb{R}_{++}} \left\{ \alpha + \beta(\varepsilon + \epsilon) + \sum_{c=1}^{C} \sup_{q_c \in \mathbb{R}_+} q_c(t_c - \alpha + \beta \log \widehat{p}_c - \beta \rho_c - \beta \log q_c) \right\},
$$

where the last equality holds because the supremum problem is separable in each decision variable $q_c$. It now follows from the first-order optimality condition that the maximizer $q_c^\star$ is

$$
q_c^\star = \exp\Big(\frac{t_c - \alpha + \beta \log \widehat{p}_c - \beta \rho_c - \beta}{\beta}\Big) > 0,
$$

and by substituting this maximizer into the objective function, the value of the support function $h_{\mathcal{Q}_\epsilon}(t)$ is then equal to the optimal value of the below optimization problem

$$
\inf_{\alpha \in \mathbb{R}, \ \beta \in \mathbb{R}_{++}} \ \alpha + \beta(\varepsilon + \epsilon) + \beta \sum_{c=1}^{C} \widehat{p}_c \exp\Big(\frac{t_c - \alpha}{\beta} - \rho_c - 1\Big).
$$

We now proceed to the second step. Denote temporarily the objective function of the above problem as $G(\epsilon, \gamma)$, where $\gamma = [\alpha; \beta]$ combines both dual variables $\alpha$ and $\beta$. Define the function

$$
g(\epsilon) = \inf_{\gamma \in \Gamma} \ G(\epsilon, \gamma), \qquad \text{with } \Gamma \triangleq \mathbb{R} \times \mathbb{R}_{++}.
$$

Because $G$ is continuous, [11, Lemma 2.7] implies that $g$ is upper-semicontinuous at 0. Furthermore, $G$ is calm from below at $\epsilon = 0$ because $G(\epsilon, \gamma) - G(0, \gamma) = \beta\epsilon \geq 0$, thus [11, Lemma 2.7] implies that $g$ is lower-semicontinuous at 0. These two facts lead to the continuity of $g$ at 0. From the first part of the proof, we have $g(\epsilon) = h_{\mathcal{Q}_\epsilon}(t)$ for any $\epsilon \in \mathbb{R}_+$. Moreover, by applying Berge's maximum theorem [5] to (A.2a), $h_{\mathcal{Q}_\epsilon}(t)$ is a continuous function of $\epsilon$ over $\mathbb{R}_+$. Thus we find

$$
h_{\mathcal{Q}}(t) = h_{\mathcal{Q}_0}(t) = \lim_{\epsilon \downarrow 0} h_{\mathcal{Q}_\epsilon}(t) = \lim_{\epsilon \downarrow 0} g(\epsilon) = g(0),
$$

where the chain of equalities follows from the definition of $\mathcal{Q}_\epsilon$, the continuity of $h_{\mathcal{Q}_\epsilon}(t)$ in $\epsilon$, the fact that $g(\epsilon) = h_{\mathcal{Q}_\epsilon}(t)$ for $\epsilon > 0$, and the continuity of $g$ at 0 established previously. The proof is now completed. □

*Proof of Proposition 3.1.* To facilitate the proof, we define the following ambiguity set over the marginal distribution of the covariate $X$ as

$$\mathbb{B}_X \triangleq \left\{ \mathbb{Q}_X \in \mathcal{M}(\mathcal{X}) : \mathrm{KL}(\mathbb{Q}_X \parallel \widehat{\mathbb{P}}_X) + \mathbb{E}_{\mathbb{Q}_X}[\sum_{c=1}^{C} \rho_c \mathbb{1}_{\widehat{x}_c}(X)] \le \varepsilon \right\}.$$

Given a nominal marginal distribution $\widehat{\mathbb{P}}_X$ supported on a finite set $\{\widehat{x}_c\}_{c\in\mathcal{C}}$, the absolute continuity requirement suggests that $\mathrm{KL}(\mathbb{Q}_X \parallel \widehat{\mathbb{P}}_X)$ is finite if and only if $\mathbb{Q}_X$ is absolutely continuous with respect to $\widehat{\mathbb{P}}_X$. Thus, any $\mathbb{Q}_X$ of interest should be supported on the same set $\{\widehat{x}_c\}_{c=1,\dots,C}$, and $\mathbb{Q}_X$ and be finitely parametrized by a $C$-dimensional vector $\{q_c\}_{c=1,\dots,C}$. Let $\mathcal{Q}$ denote the convex compact feasible set in $\mathbb{R}^C$, that is,

$$\mathcal{Q} \triangleq \left\{ q \in \mathbb{R}_+^C : \sum_{c=1}^{C} q_c = 1, \sum_{c=1}^{C} q_c(\log q_c - \log \widehat{p}_c + \rho_c) \le \varepsilon \right\},$$

and the ambiguity set $\mathbb{B}_X$ can now be finitely parametrized as

$$\mathbb{B}_X = \left\{ \mathbb{Q}_X \in \mathcal{M}(\mathcal{X}) : \exists q \in \mathcal{Q}, \ \mathbb{Q}_X = \sum_{i=1}^{C} q_c \delta_{\widehat{x}_c} \right\}.$$

By coupling $\mathbb{B}_X$ with the conditional ambiguity sets $\mathbb{B}_{Y|\widehat{x}_c}$, $\mathbb{B}(\widehat{\mathbb{P}})$ can be re-written as

$$\mathbb{B}(\widehat{\mathbb{P}}) = \left\{ \mathbb{Q} \in \mathcal{M}(\mathcal{X}\times\mathcal{Y}) : \begin{array}{l} \exists \mathbb{Q}_X \in \mathbb{B}_X, \ \mathbb{Q}_{Y|\widehat{x}_c} \in \mathbb{B}_{Y|\widehat{x}_c} \quad \forall c = 1,\dots,C \\ \mathbb{Q}(\{\widehat{x}_c\}\times A) = \widehat{\mathbb{Q}}_X(\{\widehat{x}_c\})\mathbb{Q}_{Y|\widehat{x}_c}(A) \,\forall A \in \mathcal{F}(\mathcal{Y}) \quad \forall c = 1,\dots,C \end{array} \right\}$$

The worst-case expected loss becomes

$$\sup_{\mathbb{Q}\in\mathbb{B}(\widehat{\mathbb{P}})} \mathbb{E}_{\mathbb{Q}}[L(X,Y)] = \sup_{\mathbb{Q}_X\in\mathbb{B}_X} \mathbb{E}_{\mathbb{Q}_X}\left[ \sup_{\mathbb{Q}_{Y|X}\in\mathbb{B}_{Y|X}} \mathbb{E}_{\mathbb{Q}_{Y|X}}[L(X,Y)] \right]$$

$$= \sup_{q\in\mathcal{Q}} \sum_{c=1}^{C} q_c \sup_{\mathbb{Q}_{Y|\widehat{x}_c}\in\mathbb{B}_{Y|\widehat{x}_c}} \mathbb{E}_{\mathbb{Q}_{Y|\widehat{x}_c}}[L(\widehat{x}_c,Y)],$$

where the first equality follows from the law of total expectation, and the second equality follows from the finite reparametrization of $\mathbb{B}_X$. If we denote by $\mathcal{T}$ the epigraph reformulation of the worst-case conditional expectations

$$\mathcal{T} \triangleq \left\{ t \in \mathbb{R}^C : \sup_{\mathbb{Q}_{Y|\widehat{x}_c}\in\mathbb{B}_{Y|\widehat{x}_c}} \mathbb{E}_{\mathbb{Q}_{Y|\widehat{x}_c}}[L(\widehat{x}_c,Y)] \le t_c \quad \forall c = 1,\dots,C \right\},$$

then the worst-case expected loss can be further re-expressed as

$$\sup_{\mathbb{Q}\in\mathbb{B}(\widehat{\mathbb{P}})} \mathbb{E}_{\mathbb{Q}}[L(X,Y)] = \sup_{q\in\mathcal{Q}} \inf_{t\in\mathcal{T}} q^\top t \tag{A.3a}$$

$$= \inf_{t\in\mathcal{T}} \sup_{q\in\mathcal{Q}} q^\top t \tag{A.3b}$$

$$= \begin{cases} \inf & \alpha + \beta\varepsilon + \beta \sum_{c=1}^{C} \widehat{p}_c \exp\left( \frac{t_c - \alpha}{\beta} - \rho_c - 1 \right) \\ \text{s.t.} & t \in \mathcal{T}, \ \alpha \in \mathbb{R}, \ \beta \in \mathbb{R}_{++}, \end{cases} \tag{A.3c}$$

where the sup-inf formulation (A.3a) is justified because $q$ is non-negative and we can resort to the epigraph formulations of the worst-case conditional expected loss. In (A.3b) we applied Sion's minimax theorem [13], which is valid because the sup-inf program (A.3a) is a concave-convex saddle problem, and $\mathcal{Q}$ is convex and compact and $\mathcal{T}$ is convex. In (A.3c) we have used Lemma B.1 to reformulate the supremum over $q$. The claim then follows. $\square$

Instead of solving the problem in the natural parameters $\theta$ coupled with its log-partition function $\Psi$, we will use the reparametrization to the mean parameters using the conjugate function of $\Psi$. More specifically, let $\phi$ be the convex conjugate of $\Psi$, that is,

$$\phi : \mu \mapsto \sup_{\theta\in\Theta} \left\{ \langle \mu, \theta \rangle - \Psi(\theta) \right\}$$

Before proceeding to the technical proofs, the below lemma collects from the existing literature the necessary background knowledge about the log-partition function $\Psi$ and its conjugate $\phi$, along with the relationship between the natural parameter $\theta$ and its corresponding expectation parameter $\mu$.

**Lemma B.2** (Relevant facts)**.** The following assertions hold for regular exponential family.

(i) The function $\phi$ is closed, convex and proper on $\mathbb{R}^p$.

(ii) $(\Theta, \Psi)$ and $(\mathrm{int}(\mathrm{dom}(\phi)), \phi)$ are convex functions of Legendre type, and they are Legendre duals of each other.

(iii) The gradient function $\nabla \Psi$ is a one-to-one function from the open convex set $\Theta$ onto the open convex set $\mathrm{int}(\mathrm{dom}(\phi))$.

(iv) The gradient functions $\nabla \Psi$ and $\nabla \phi$ are continuous, and $\nabla \phi = (\nabla \Psi)^{-1}$.

(v) The function $\phi$ is essentially smooth over $\mathrm{int}(\mathrm{dom}(\phi))$.

*Proof of Lemma B.2.* Assertion (i) holds since $\langle \mu, \theta \rangle - \Psi(\theta)$ is convex and closed for each $\theta$, thus taking supremum, $\phi$ is convex and closed. $\phi$ is proper since $\mathrm{dom}(\phi)$ is non-empty. Assertions (ii) to (iv) follows from [3, Lemma 1] and [3, Theorem 2]. Assertion (v) follows from [3, Lemma 1] and [12, Theorem 26.3], and the fact that $\Psi$ and $\phi$ is a convex conjugate pair. $\square$

From Assertion (ii), we have the mappings between the dual spaces $\mathrm{int}(\mathrm{dom}(\phi))$ and $\Theta$ are given by the Legendre transformation

$$\mu(\theta) = \nabla \Psi(\theta) \quad \text{and} \quad \theta(\mu) = \nabla \phi(\mu).$$

For any $\mu \in \mathrm{int}(\mathrm{dom}(\phi))$, the conjugate function $\phi$ can be expressed as

$$\phi(\mu) = \langle \mu, \theta(\mu) \rangle - \Psi(\theta(\mu)).$$

**Lemma B.3** (KL divergence between distributions from exponential family)**.** Suppose that $\mathbb{Q}_1$ and $\mathbb{Q}_2$ belong to the exponential family of distributions with the same log-partition function $\Psi$ and with natural parameters $\theta_1$ and $\theta_2$ respectively. The KL divergence from $\mathbb{Q}_1$ to $\mathbb{Q}_2$ amounts to

$$\mathrm{KL}(\mathbb{Q}_1 \parallel \mathbb{Q}_2) = \langle \theta_1 - \theta_2, \mu_1 \rangle - \Psi(\theta_1) + \Psi(\theta_2) = \phi(\mu_1) - \phi(\mu_2) - \langle \mu_1 - \mu_2, \theta_2 \rangle,$$

where $\phi$ is the convex conjugate of $\Psi$, and $\mu_j = \nabla \Psi(\theta_j)$ for any $j \in \{1, 2\}$.

The result of Lemma B.3 can be found in [3, Appendix A], but the explicit proof is included here for completeness.

*Proof of Lemma B.3.* One finds

$$\mathrm{KL}(\mathbb{Q}_1 \parallel \mathbb{Q}_2) = \mathbb{E}_{\mathbb{Q}_1}[\log(\mathrm{d}\mathbb{Q}_1/\mathrm{d}\mathbb{Q}_2)]$$
$$= \mathbb{E}_{\mathbb{Q}_1}[\langle T(Y), \theta_1 - \theta_2 \rangle - \Psi(\theta_1) + \Psi(\theta_2)] \tag{A.4a}$$
$$= \langle \mu_1, \theta_1 - \theta_2 \rangle - \Psi(\theta_1) + \Psi(\theta_2), \tag{A.4b}$$

where equality (A.4a) follows by calculating the logarithm of the Radon-Nikodym derivatives between two distributions, and equality (A.4b) follows by noting that $\mu_1 = \mathbb{E}_{\mathbb{Q}_1}[T(Y)]$.

By [3, Theorem 4], one can also rewrite the density using the mean parameter $\mu = \mu(\theta)$ as

$$f(y|\mu) = h(y) \exp\left( \langle \theta, T(y) \rangle - \Psi(\theta) \right)$$
$$= h(y) \exp\left( \phi(\mu) + \langle T(y) - \mu, \nabla \phi(\mu) \rangle \right)$$

The KL divergence from $\mathbb{Q}_1$ to $\mathbb{Q}_2$ amounts to

$$\mathrm{KL}(\mathbb{Q}_1 \parallel \mathbb{Q}_2) = \mathbb{E}_{\mathbb{Q}_1}[\log(\mathrm{d}\mathbb{Q}_1/\mathrm{d}\mathbb{Q}_2)]$$
$$= \mathbb{E}_{\mathbb{Q}_1}[\phi(\mu_1) - \phi(\mu_2) + \langle T(Y), \nabla \phi(\mu_1) - \nabla \phi(\mu_2) \rangle - \langle \mu_1, \nabla \phi(\mu_1) \rangle + \langle \mu_2, \nabla \phi(\mu_2) \rangle]$$
$$\tag{A.5a}$$
$$= \langle \mu_2 - \mu_1, \theta_2 \rangle + \phi(\mu_1) - \phi(\mu_2). \tag{A.5b}$$

From Assertion (iv) in Lemma B.2, we notice that $\theta_2 = \nabla \phi(\mu_2)$, which completes the proof. $\square$

Recall that the conditional ambiguity set defined in (8) is

$$\mathbb{B}_{Y|\widehat{x}_c} \triangleq \left\{ \mathbb{Q}_{Y|\widehat{x}_c} \in \mathcal{M}(\mathcal{Y}) : \exists \theta \in \Theta, \ \mathbb{Q}_{Y|\widehat{x}_c}(\cdot) \sim f(\cdot|\theta), \ \mathrm{KL}(\mathbb{Q}_{Y|\widehat{x}_c} \parallel \widehat{\mathbb{P}}_{Y|\widehat{x}_c}) \leq \rho_c \right\}$$

for a parametric, nominal conditional measure $\widehat{\mathbb{P}}_{Y|\widehat{x}_c} \sim f(\cdot|\widehat{\theta}_c), \widehat{\theta}_c \in \Theta$ and a radius $\rho_c \in \mathbb{R}_+$. The uncertainty set $\mathcal{S}_c$ of expectation parameters induced by the ambiguity set $\mathbb{B}_{Y|\widehat{x}_c}$ is defined as

$$\mathcal{S}_c \triangleq \left\{ \mu \in \mathrm{dom}(\phi) : \exists \mathbb{Q}_{Y|\widehat{x}_c} \in \mathbb{B}_{Y|\widehat{x}_c}, \ \mu = \mathbb{E}_{\mathbb{Q}_{Y|\widehat{x}_c}}[T(Y)] \right\}.$$

**Lemma B.4** (Compactness of expectation parameter uncertainty set). The set $\mathcal{S}_c$ is compact, and it has an interior point whenever $\rho_c > 0$.

*Proof of Lemma B.4.* By Lemma B.3 and the definition of the set $\mathcal{S}_c$, we can write $\mathcal{S}_c$ as

$$\mathcal{S}_c = \left\{ \mu \in \mathrm{dom}(\phi) : \phi(\mu) - \phi(\widehat{\mu}_c) - \langle \mu - \widehat{\mu}_c, \widehat{\theta}_c \rangle \leq \rho_c \right\}.$$

Because $\phi$ is closed, convex, proper, and that $\widehat{\theta}_c \in \mathrm{int}(\Theta) = \Theta$, the function $\phi(\cdot) - \langle \cdot, \widehat{\theta}_c \rangle$ is coercive by [12, Corollary 14.2.2] and [4, Fact 2.11]. As a consequence, $\mathcal{S}_c$ is bounded.

Because $\Psi$ is essentially strictly convex on $\Theta$, $\phi$ is essentially smooth on $\mathrm{int}(\mathrm{dom}(\phi))$ by [12, Theorem 26.3]. [4, Theorem 3.8] now implies that if $\mu'$ is a boundary point of $\mathrm{int}(\mathrm{dom}(\phi))$ then as $\mathrm{int}(\mathrm{dom}(\phi)) \ni \mu_k \xrightarrow{k \to \infty} \mu'$ then $\phi(\mu_k) - \langle \mu_k, \widehat{\theta}_c \rangle \xrightarrow{k \to \infty} +\infty$. Moreover, because $\phi$ is continuous over $\mathrm{int}(\mathrm{dom}(\phi))$, the set $\mathcal{S}_c$ is closed. This implies that $\mathcal{S}_c$, being a closed and bounded set of finite dimension, is compact.

The continuity of $\phi$ leads a straightforward manner to the non-empty interior of $\mathcal{S}_c$ when $\rho_c > 0$. This observation completes the proof. □

*Proof of Proposition 3.2.* Because $\lambda$ is a mapping onto the space $\Theta$ of natural parameters, we use the shorthand $\lambda_c = \lambda(w, \widehat{x}_c) \in \Theta$. Moreover, let $\widehat{\mu}_c = \nabla\Psi(\widehat{\theta}_c)$. The worst-case conditional expectation of the log-loss function becomes

$$\begin{aligned}
\sup_{\mathbb{Q}_{Y|\widehat{x}_c} \in \mathbb{B}_{Y|\widehat{x}_c}} \mathbb{E}_{\mathbb{Q}_{Y|\widehat{x}_c}}[\ell_\lambda(\widehat{x}_c, Y, w)] &= \sup_{\mathbb{Q}_{Y|\widehat{x}_c} \in \mathbb{B}_{Y|\widehat{x}_c}} \mathbb{E}_{\mathbb{Q}_{Y|\widehat{x}_c}}\left[ \Psi(\lambda(w, \widehat{x}_c)) - \langle T(Y), \lambda(w, \widehat{x}_c) \rangle \right] \\
&= \sup_{\mathbb{Q}_{Y|\widehat{x}_c} \in \mathbb{B}_{Y|\widehat{x}_c}} \Psi(\lambda(w, \widehat{x}_c)) - \langle \mathbb{E}_{\mathbb{Q}_{Y|\widehat{x}_c}}[T(Y)], \lambda(w, \widehat{x}_c) \rangle \\
&= \begin{cases} \sup & \Psi(\lambda_c) - \langle \mu, \lambda_c \rangle \\ \mathrm{s.\,t.} & \phi(\mu) - \phi(\widehat{\mu}_c) - \langle \mu - \widehat{\mu}_c, \widehat{\theta}_c \rangle \leq \rho_c, \end{cases}
\end{aligned}$$

where the first equality is from the definition of $\ell_\lambda$ and the second equality follows from the linearity of the expectation operator. The last equality follows from the definition of the ambiguity set $\mathbb{B}_{Y|\widehat{x}_c}$ using the $\phi$ function by Lemma B.3. Because the term $\Psi(\lambda_c)$ does not involve the decision variable $\mu$, it suffices now to consider the optimization problem

$$\sup \left\{ \langle -\lambda_c, \mu \rangle \ : \ \phi(\mu) - \langle \mu, \widehat{\theta}_c \rangle \leq \rho_c + \phi(\widehat{\mu}_c) - \langle \widehat{\mu}_c, \widehat{\theta}_c \rangle \right\}. \tag{A.7}$$

Suppose at this moment that $\lambda_c \neq 0$ and $\rho_c > 0$. When $\rho_c > 0$, the feasible set of (A.7) satisfies the Slater condition because $\phi$ is a continuous function. Hence, by a strong duality argument, the convex optimization problem (A.7) is equivalent to

$$\sup_\mu \inf_{\gamma \geq 0} \langle -\lambda_c, \mu \rangle + \gamma(\bar{\rho}_c - \phi(\mu) + \langle \mu, \widehat{\theta}_c \rangle) = \inf_{\gamma \geq 0} \left\{ \gamma\bar{\rho}_c + \sup_\mu \langle \mu, \gamma\widehat{\theta}_c - \lambda_c \rangle - \gamma\phi(\mu) \right\},$$

where $\bar{\rho}_c \triangleq \rho_c + \phi(\widehat{\mu}_c) - \langle \widehat{\mu}_c, \widehat{\theta}_c \rangle \in \mathbb{R}$ and the interchange of the supremum and the infimum operators is justified thanks to [6, Proposition 5.3.1]. Consider now the infimum problem on the right hand side of the above equation. If $\gamma = 0$, then the inner supremum subproblem on the right hand side is unbounded because $\lambda_c \neq 0$, thus $\gamma = 0$ is never an optimal solution to the infimum problem. By utilizing the definition of the conjugate function, one thus deduce that problem (A.7) is equivalent to

$$\inf_{\gamma > 0} \gamma\bar{\rho}_c + (\gamma\phi)^*(\gamma\widehat{\theta}_c - \lambda_c) = \inf_{\gamma > 0} \gamma\bar{\rho}_c + \gamma\phi^*\left( \widehat{\theta}_c - \frac{\lambda_c}{\gamma} \right), \tag{A.8}$$

where the equality exploits the fact that $(\gamma\phi)^*(\theta) = \gamma\phi^*(\theta/\gamma)$ for any $\gamma > 0$ [7, Table 3.2].

We now show that the reformulation problem (A.8) is valid when $\rho_c = 0$. Indeed, when $\rho_c = 0$, problem (A.7) has a unique feasible solution $\widehat{\mu}_c$, thus its optimal value is $\langle -\lambda_c, \widehat{\mu}_c \rangle$. Moreover, in this case, problem (A.8) becomes

$$\inf_{\gamma > 0} \gamma \left[ \phi(\widehat{\mu}_c) - \langle \widehat{\mu}_c, \widehat{\theta}_c \rangle + \phi^* \left( \widehat{\theta}_c - \frac{\lambda_c}{\gamma} \right) \right]$$

$$= \langle -\lambda_c, \widehat{\mu}_c \rangle + \inf_{\gamma > 0} \gamma \left[ \phi(\widehat{\mu}_c) - \langle \widehat{\mu}_c, \widehat{\theta}_c - \frac{\lambda_c}{\gamma} \rangle + \phi^* \left( \widehat{\theta}_c - \frac{\lambda_c}{\gamma} \right) \right].$$

Notice that the term in the square bracket of the optimization problem on the right hand side is non-negative by the definition of the conjugate function. Thus, the infimum problem over $\gamma$ admits the optimal value of 0 as $\gamma$ tends to $+\infty$. As a consequence, when $\rho_c = 0$, both problem (A.7) and (A.8) have the same optimal value and they are equivalent.

Consider now the situation where $\lambda_c = 0$. In this case, problem (A.8) becomes

$$\inf_{\gamma > 0} \gamma\rho_c + \gamma \left( \phi(\widehat{\mu}_c) - \langle \widehat{\mu}_c, \widehat{\theta}_c \rangle + \phi^*(\widehat{\theta}_c) \right).$$

By definition of the conjugate function, we have $\phi^*(\widehat{\theta}_c) \geq \langle \widehat{\mu}_c, \widehat{\theta}_c \rangle - \phi(\widehat{\mu}_c)$, and thus, by combining with the fact that $\rho_c \geq 0$, this infimum problem will admit the optimal value of 0. Notice that when $\lambda_c = 0$, the optimal value of problem (A.7) is also 0. This shows that (A.8) is equivalent to (A.7) for any possible value of $\lambda_c$. Replacing $\phi^*$ in (A.8) by its equivalence $\Psi$ and substituting $\langle \widehat{\mu}_c, \widehat{\theta}_c \rangle - \phi(\widehat{\mu}_c)$ by its equivalence $\Psi(\widehat{\theta}_c)$ complete the reformulation (10). □

*Proof of Theorem 3.3.* By applying Proposition 3.1, the distributionally robust MLE problem (4) can be reformulated as

$$\min_{w \in \mathcal{W}} \max_{\mathbb{Q} \in \mathbb{B}(\widehat{\mathbb{P}})} \mathbb{E}_{\mathbb{Q}} \left[ \ell_\lambda(X, Y, w) \right] = \begin{cases} \inf & \alpha + \beta\varepsilon + \beta \sum_{c=1}^{C} \widehat{p}_c \exp \left( \frac{t_c - \alpha}{\beta} - \rho_c - 1 \right) \\ \text{s.t.} & w \in \mathcal{W}, \ t \in \mathbb{R}^C, \ \alpha \in \mathbb{R}, \ \beta \in \mathbb{R}_{++} \\ & \sup_{\mathbb{Q}_{Y|\widehat{x}_c} \in \mathbb{B}_{Y|\widehat{x}_c}} \mathbb{E}_{\mathbb{Q}_{Y|\widehat{x}_c}} \left[ \ell_\lambda(\widehat{x}_c, Y, w) \right] \leq t_c \ \ \forall c = 1, \dots, C. \end{cases}$$

Using Proposition 3.2 to reformulate each constraint of the above optimization problem leads to the desired result. □

## C Proofs of Section 4

*Proof of Proposition 4.1.* Let $\mathbb{1}$ denote the $N$ dimensional vector of all 1's. Let $\text{KL}(q \parallel p) = \sum_{i=1}^{N} q_i \log(q_i/p_i)$, we have

$$\sup_{\mathbb{Q}:\text{KL}(\mathbb{Q}\|\widehat{\mathbb{P}}^{\text{emp}}) \leq \varepsilon} \mathbb{E}_{\mathbb{Q}}[\ell_\lambda(X, Y, w)] = \sup_{q:\text{KL}(q\|\frac{1}{N}\mathbb{1}) \leq \varepsilon} \sum_{i=1}^{N} q_i \ell_\lambda(\widehat{x}_i, \widehat{y}_i, w)$$

$$= \sup_{q:\text{KL}(q\|\frac{1}{N}\mathbb{1}) \leq \varepsilon} \sum_{i=1}^{N} q_i \left( \Psi(\lambda(w, \widehat{x}_i)) - \langle T(\widehat{y}_i), \lambda(w, \widehat{x}_i) \rangle \right).$$

On the other hand, we note

$$\sup_{\mathbb{Q} \in \mathbb{B}(\widehat{\mathbb{P}})} \mathbb{E}_{\mathbb{Q}} \left[ \ell_\lambda(X, Y, w) \right] = \sup_{q:\text{KL}(q\|\frac{1}{N}\mathbb{1}) \leq \varepsilon} \sum_{i=1}^{N} q_i \mathbb{E}_{\mathbb{Q}_{Y|\widehat{x}_i}} \left[ \ell_\lambda(\widehat{x}_i, Y, w) \right]$$

$$= \sup_{q:\text{KL}(q\|\frac{1}{N}\mathbb{1}) \leq \varepsilon} \sum_{i=1}^{N} q_i \left( \Psi(\lambda(w, \widehat{x}_i)) - \langle \nabla\Psi(\widehat{\theta}_i), \lambda(w, \widehat{x}_i) \rangle \right)$$

$$= \sup_{q:\text{KL}(q\|\frac{1}{N}\mathbb{1}) \leq \varepsilon} \sum_{i=1}^{N} q_i \left( \Psi(\lambda(w, \widehat{x}_i)) - \langle T(\widehat{y}_i), \lambda(w, \widehat{x}_i) \rangle \right).$$

Therefore the objective functions are the same and the two problems are equivalent. □

The proof of Proposition 4.2 relies on the following result.

**Lemma C.1.** Let $\Delta \subset \mathbb{R}^C$ be a simplex and $\widehat{p} \in \text{int}(\Delta)$ be a probability vector. For any two vectors $\widehat{t}, t^\star \in \mathbb{R}^C$, any vector $\rho \in \mathbb{R}_+^C$ and any scalar $\varepsilon \geq \widehat{p}^\top \rho$, we have

$$\sup\left\{ q^\top t^\star - \widehat{p}^\top \widehat{t} : q \in \Delta, \sum_{c=1}^C q_c(\log q_c - \log \widehat{p}_c + \rho_c) \leq \varepsilon \right\}$$

$$\leq \|t^\star - \widehat{t}\|_\infty + \frac{\sqrt{2\varepsilon}}{\min_c \sqrt{\widehat{p}_c}} \sqrt{\sum_{c=1}^C \widehat{p}_c(\widehat{t}_c - \bar{t})^2},$$

where $\bar{t} = \widehat{p}^\top \widehat{t}$.

*Proof of Lemma C.1.* Let $\mathbb{1}$ denote the $C$ dimensional vector of 1's, we have

$$\begin{cases} \sup & q^\top t^\star - \widehat{p}^\top \widehat{t} \\ \text{s.t.} & q \in \Delta, \ \sum_{c=1}^C q_c(\log q_c - \log \widehat{p}_c + \rho_c) \leq \varepsilon \end{cases}$$

$$= \begin{cases} \sup & q^\top(t^\star - \widehat{t}) + (q - \widehat{p})^\top \widehat{t} \\ \text{s.t.} & q \in \Delta, \ \sum_{c=1}^C q_c(\log q_c - \log \widehat{p}_c + \rho_c) \leq \varepsilon \end{cases}$$

$$\leq \begin{cases} \sup & q^\top(t^\star - \widehat{t}) + (q - \widehat{p})^\top \widehat{t} \\ \text{s.t.} & q \in \Delta, \ \sum_{c=1}^C (q_c - \widehat{p}_c)^2 \leq 2\varepsilon \end{cases}$$

$$\leq \sup_{\|q\|_1=1} q^\top(t^\star - \widehat{t}) + \sup\left\{ (q - \widehat{p})^\top(\widehat{t} - \bar{t}\mathbb{1}) : \|q - \widehat{p}\|_2^2 \leq 2\varepsilon \right\}$$

$$\leq \sup_{\|q\|_1=1} q^\top(t^\star - \widehat{t}) + \sup\left\{ \sum_{c=1}^C \frac{q_c - \widehat{p}_c}{\sqrt{\widehat{p}_c}} \sqrt{\widehat{p}_c}(\widehat{t}_c - \bar{t}) : \|q - \widehat{p}\|_2^2 \leq 2\varepsilon \right\}$$

$$\leq \sup_{\|q\|_1=1} q^\top(t^\star - \widehat{t}) + \frac{\sqrt{2\varepsilon}}{\min_c \sqrt{\widehat{p}_c}} \sqrt{\sum_{c=1}^C \widehat{p}_c(\widehat{t}_c - \bar{t})^2},$$

where the first inequality follows from Pinsker's inequality [8, Theorem 4.19] and the fact that $\|q - \widehat{p}\|_2^2 \leq \|q - \widehat{p}\|_1^2 = 4\|q - \widehat{p}\|_{TV}^2$, the second inequality follows from the fact that $(q - \widehat{p})^\top \mathbb{1} = 0$ and dropping the constraint $q \in \Delta$, and the last inequality is from Cauchy-Schwarz.

In the last step, we have

$$\sup_{\|q\|_1=1} q^\top(t^\star - \widehat{t}) = \|t^\star - \widehat{t}\|_\infty,$$

which completes the proof. $\square$

We now ready to prove Proposition 4.2.

*Proof of Proposition 4.2.* Let $t^\star$ and $\widehat{t}$ be two $C$-dimensional vectors whose elements are defined as

$$t_c^\star = \sup_{\mathbb{Q}_{Y|\widehat{x}_c} \in \mathbb{B}_{Y|\widehat{x}_c}} \mathbb{E}_{\mathbb{Q}_{Y|\widehat{x}_c}}[\ell_\lambda(\widehat{x}_c, Y, w)], \quad \widehat{t}_c = \mathbb{E}_{\widehat{\mathbb{P}}_{Y|\widehat{x}_c}}[\ell_\lambda(\widehat{x}_c, Y, w)] \qquad \forall c.$$

By Lemma C.1, we find

$$\sup_{\mathbb{Q} \in \mathbb{B}(\widehat{\mathbb{P}})} \mathbb{E}_{\mathbb{Q}}[\ell_\lambda(X, Y, w)] - \mathbb{E}_{\widehat{\mathbb{P}}}[\ell_\lambda(X, Y, w)] = \begin{cases} \sup & q^\top t^\star - \widehat{p}^\top \widehat{t} \\ \text{s.t.} & q \in \Delta, \ \sum_{c=1}^C q_c(\log q_c - \log \widehat{p}_c + \rho_c) \leq \varepsilon \end{cases}$$

$$\leq \|t^\star - \widehat{t}\|_\infty + \frac{\sqrt{2\varepsilon}}{\min_c \sqrt{\widehat{p}_c}} \sqrt{\sum_{c=1}^C \widehat{p}_c(\widehat{t}_c - \bar{t})^2},$$

where $\bar{t} = \widehat{p}^\top \widehat{t}$. In the last step, notice that

$$\sum_{c=1}^{C} \widehat{p}_c (\widehat{t}_c - \bar{t})^2 = \operatorname{Var}_{\widehat{\mathbb{P}}_X} \left( \mathbb{E}_{\widehat{\mathbb{P}}_{Y|X}} [\ell_\lambda(X, Y, w)] \right) \leq \operatorname{Var}_{\widehat{\mathbb{P}}} (\ell_\lambda(X, Y, w)) .$$

It now remains to provide the bounds for $\|t^\star - \widehat{t}\|_\infty$. For any $c$, let $\lambda_c = \lambda(w, \widehat{x}_c)$, we have

$$t_c^\star - \widehat{t}_c = \begin{cases} \sup & \langle \mu - \widehat{\mu}_c, \lambda_c \rangle \\ \text{s.t.} & \phi(\mu) - \phi(\widehat{\mu}_c) - \langle \mu - \widehat{\mu}_c, \widehat{\theta}_c \rangle \leq \rho_c. \end{cases}$$

Because $\Psi$ has locally Lipschitz continuous gradients, $\phi$ is locally strongly convex [9, Theorem 4.1]. Moreover, the feasible set $\mathcal{S}_c$ of the above problem is compact by Lemma B.4, hence there exists a constant $0 < m_c$ such that

$$\frac{m_c}{2} \|\mu - \widehat{\mu}_c\|_2^2 \leq \phi(\mu) - \phi(\widehat{\mu}_c) - \langle \mu - \widehat{\mu}_c, \widehat{\theta}_c \rangle \quad \forall \mu \in \mathcal{S}_c.$$

Notice that the constants $m_c$ depends only on $\Psi$ and $\widehat{\theta}_c$. Thus, we find

$$t_c^\star - \widehat{t}_c \leq \sup \left\{ \langle \mu - \widehat{\mu}_c, \lambda_c \rangle : m_c \|\mu - \widehat{\mu}_c\|_2^2 \leq 2\rho_c \right\} = \sqrt{2\rho_c/m_c} \|\lambda(w, \widehat{x}_c)\|_2.$$

By setting $m = \min_c m_c$, we have

$$\|t^\star - \widehat{t}\|_\infty \leq \sqrt{\frac{2 \max_c \rho_c}{m}} \|\lambda(w, \widehat{x}_c)\|_2.$$

Combining terms leads to the postulated results. $\qquad\qquad\qquad\qquad\qquad\qquad\qquad\qquad\qquad\qquad \square$

For any $\widehat{\theta}_c \in \Theta$, $\rho_c \in \mathbb{R}_+$, let $\mathcal{R}_{\widehat{\theta}_c, \rho_c}(w)$ denote the value of the worst-case expected log-loss

$$\mathcal{R}_{\widehat{\theta}_c, \rho_c}(w) = \sup_{\mathbb{Q}_{Y|\widehat{x}_c} \in \mathbb{B}_{Y|\widehat{x}_c}} \mathbb{E}_{\mathbb{Q}_{Y|\widehat{x}_c}} [\ell_\lambda(\widehat{x}_c, Y, w)] .$$

**Lemma C.2.** Suppose that the log-partition function $\Psi$ has locally Lipschitz continuous gradients, that $\Theta = \mathbb{R}^p$ and that $\Theta_c \subset \Theta$ is a compact set. For any fixed $\overline{\rho}_c \in \mathbb{R}_{++}$, there exist constants $0 < m < M < +\infty$ that depend only on $\Psi$, $\Theta_c$ and $\overline{\rho}_c$ such that for any value $\lambda(w, \widehat{x}_c) \in \mathbb{R}^p$ and any radius $\overline{\rho}_c \geq \rho_c \geq 0$

$$\sqrt{2\rho_c/M} \|\lambda(w, \widehat{x}_c)\|_2 \leq \mathcal{R}_{\widehat{\theta}_c, \rho_c}(w) - \mathcal{R}_{\widehat{\theta}_c, 0}(w) \leq \sqrt{2\rho_c/m} \|\lambda(w, \widehat{x}_c)\|_2 \qquad \forall \widehat{\theta}_c \in \Theta_c.$$

*Proof of Lemma C.2.* Consider the set

$$\mathcal{D} \triangleq \{ \widehat{\mu}_c : \exists \widehat{\theta}_c \in \Theta_c \text{ such that } \widehat{\mu}_c = \nabla \Psi(\widehat{\theta}_c) \}$$

and its $\overline{\rho}_c$-inflated set

$$\mathcal{D}_{\overline{\rho}_c} \triangleq \{ \mu : \exists \widehat{\mu}_c \in \mathcal{D} \text{ such that } \phi(\mu) - \phi(\widehat{\mu}_c) - \langle \mu - \widehat{\mu}_c, \widehat{\theta}_c \rangle \leq \overline{\rho}_c \}.$$

Because $\Theta_c$ is compact and $\nabla \Psi$ is a continuous function, $\mathcal{D}$ is compact [1, Theorem 2.34]. Note that we can rewrite $\mathcal{D}_{\overline{\rho}_c}$ as

$$\mathcal{D}_{\overline{\rho}_c} = \{ \mu : \exists \widehat{\mu}_c \in \mathcal{D} \text{ such that } \phi(\mu) + \langle \mu, -\widehat{\theta}_c \rangle \leq \overline{\rho}_c + \phi(\widehat{\mu}_c) - \langle \widehat{\mu}_c, \widehat{\theta}_c \rangle \}.$$

Let $S$ be temporarily the set

$$S = \left\{ \mu : \phi(\mu) + \inf_{\widehat{\theta}_c \in \Theta_c} \langle \mu, -\widehat{\theta}_c \rangle \leq \overline{\rho}_c + \sup_{\widehat{\theta}_c \in \Theta_c} \phi(\widehat{\mu}_c) - \langle \widehat{\mu}_c, \widehat{\theta}_c \rangle < \infty \right\}.$$

We have that $\mathcal{D}_{\overline{\rho}_c} \subseteq S$. Recall the definition of $\phi$:

$$\phi : \mu \mapsto \sup_{\theta \in \Theta} \left\{ \langle \mu, \theta \rangle - \Psi(\theta) \right\}.$$

Therefore $\phi(\cdot)$ is closed, convex and proper. Therefore by [4, Proposition 2.16], $\Theta = \mathbb{R}^p$ implies that $\phi(\cdot)$ is super-coercive, i.e., $\lim_{\|\mu\|_2 \to \infty} \phi(\mu)/\|\mu\|_2 \to \infty$. Thus

$$\lim_{\|\mu\|_2 \to \infty} \phi(\mu) + \inf_{\widehat{\theta}_c \in \Theta_c} \langle \mu, -\widehat{\theta}_c \rangle \to \infty.$$

Therefore $S$ is bounded, which implies that $\mathcal{D}_{\overline{\rho}_c}$ is also bounded.

Since $\Theta_c$ is compact, there exists a subsequence $\{\widehat{\theta}_c^{k_n}\}_{n \geq 1}$ such that $\widehat{\theta}_c^{k_n} \to \widehat{\theta}_c^{\infty} \in \Theta_c$ as $n \to \infty$. Since $\mathcal{D}_{\rho_c}$ is bounded, it suffices to show that $\mathcal{D}_{\rho_c}$ is closed. Choose any sequence $\{\mu^k\}_{k \geq 1} \in \mathcal{D}_{\rho_c}$ such that $\mu^k \to \mu^{\infty}$ as $k \to \infty$, we want to show that $\mu^{\infty} \in \mathcal{D}_{\rho_c}$. For each $k$, since $\mu^k \in \mathcal{D}_{\rho_c}$, there exists $\widehat{\mu}_c^k \in \mathcal{D}$ and $\widehat{\theta}_c^k \in \Theta_c$ such that $\phi(\mu^k) - \phi(\widehat{\mu}_c^k) - \langle \mu^k - \widehat{\mu}_c^k, \widehat{\theta}_c^k \rangle \leq \rho_c$. Since $\mathcal{D}$ and $\Theta_c$ are compact, there exists a subsequence $\{k_n\}_{n \geq 1}$ such that $\widehat{\mu}_c^{k_n} \to \widehat{\mu}_c^{\infty}$ and $\widehat{\theta}_c^{k_n} \to \widehat{\theta}_c^{\infty}$ for some $\widehat{\mu}_c^{\infty} \in \mathcal{D}$ and $\widehat{\theta}_c^{\infty} \in \Theta_c$. Since $\widehat{\mu}_c^{k_n} = \nabla\Psi(\widehat{\theta}_c^{k_n})$, by continuity we have $\widehat{\mu}_c^{\infty} = \nabla\Psi(\widehat{\theta}_c^{\infty})$. Note that

$$\phi(\mu^{k_n}) - \phi(\widehat{\mu}_c^{k_n}) - \langle \mu^{k_n} - \widehat{\mu}_c^{k_n}, \widehat{\theta}_c^{k_n} \rangle \leq \rho_c,$$

by continuity of $\phi$, we have

$$\phi(\mu^{\infty}) - \phi(\widehat{\mu}_c^{\infty}) - \langle \mu^{\infty} - \widehat{\mu}_c^{\infty}, \widehat{\theta}_c^{\infty} \rangle \leq \rho_c.$$

Therefore $\mu^{\infty} \in \mathcal{D}_{\rho_c}$ and hence $\mathcal{D}_{\rho_c}$ is closed.

The finite dimensional set $\mathcal{D}_{\overline{\rho}_c}$ is closed and bounded, thus it is compact, and moreover $\mathcal{D} \subseteq \mathcal{D}_{\rho_c}$. The convex hull $\overline{\mathcal{D}}_{\overline{\rho}_c}$ of $\mathcal{D}_{\overline{\rho}_c}$ is also compact [1, Corollary 5.33]. Because $\Psi$ has locally Lipschitz continuous gradients, $\phi$ is locally strongly convex [9, Theorem 4.1]. Moreover, $\phi$ is also essentially smooth by Lemma B.2(v). Thus over the set $\overline{\mathcal{D}}_{\overline{\rho}_c}$, there exist constants $0 < m \leq M < +\infty$ such that

$$\frac{m}{2}\|\mu - \mu'\|_2^2 \leq \phi(\mu) - \phi(\mu') - \langle \mu - \mu', \theta' \rangle \leq \frac{M}{2}\|\mu - \mu'\|_2^2 \quad \forall \mu, \mu' \in \overline{\mathcal{D}}_{\overline{\rho}_c}, \mu' = \nabla\Psi(\theta').$$

Notice that the constants $m$ and $M$ depend only on $\phi$, and thus on $\Psi$, $\overline{\rho}_c$ and $\Theta_c$

Denote temporarily the shorthand $\lambda_c = \lambda(w, \widehat{x}_c)$. We have $\mathcal{R}_{\widehat{\theta}_c, 0}(w) = \Psi(\lambda_c) - \langle \widehat{\mu}_c, \lambda_c \rangle$, and so

$$\mathcal{R}_{\widehat{\theta}_c, \rho_c}(w) - \mathcal{R}_{\widehat{\theta}_c, 0}(w) = \left\{ \begin{array}{ll} \sup & \langle \mu - \widehat{\mu}_c, \lambda_c \rangle \\ \text{s.t.} & \phi(\mu) - \phi(\widehat{\mu}_c) - \langle \mu - \widehat{\mu}_c, \widehat{\theta}_c \rangle \leq \rho_c. \end{array} \right.$$

Because $\mu$ and $\widehat{\mu}_c$ are both in $\overline{D}_{\overline{\rho}_c}$, we have

$$\frac{m}{2}\|\mu - \widehat{\mu}_c\|_2^2 \leq \phi(\mu) - \phi(\widehat{\mu}_c) - \langle \mu - \widehat{\mu}_c, \widehat{\theta}_c \rangle \leq \frac{M}{2}\|\mu - \widehat{\mu}_c\|_2^2.$$

We now have

$$\mathcal{R}_{\widehat{\theta}_c, \rho_c}(w) - \mathcal{R}_{\widehat{\theta}_c, 0}(w) \leq \sup\left\{\langle \mu - \widehat{\mu}_c, \lambda_c \rangle : \|\mu - \widehat{\mu}_c\|_2^2 \leq 2\rho_c/m\right\} = \sqrt{2\rho_c/m}\|\lambda_c\|_2.$$

A similar argument leads to the lower bound. This observation completes the proof. $\square$

*Proof of Theorem 4.3.* Without loss of generality consider $\mathcal{W} \subseteq \mathbb{R}^q$. For notational simplicity, denote

$$R_{\widehat{\theta}, \varepsilon, \rho}(w) = \sup_{\mathbb{Q} \in \mathbb{B}(\widehat{\mathbb{P}})} \mathbb{E}_{\mathbb{Q}}\left[\ell_\lambda(X, Y, w)\right].$$

Since $\varepsilon \geq \sum_{c=1}^C \widehat{p}_c \rho_c$ with probability going to 1, following the same argument as in the proof of Proposition 4.2, we have that with probability going to 1, for any $w \in \mathcal{W}$,

$$R_{\widehat{\theta}, \varepsilon, \rho}(w) - R_{\widehat{\theta}, 0, 0}(w) \leq \|t^\star - \widehat{t}\|_1 + \sqrt{2\varepsilon}\|\widehat{t}\|_1,$$

where

$$\|\widehat{t}\|_1 = \sum_{c=1}^C |\mathbb{E}_{\widehat{\mathbb{P}}_{Y|\widehat{x}_c}}[\ell_\lambda(\widehat{x}_c, Y, w)]| \quad \text{and} \quad \|t^\star - \widehat{t}\|_1 = \sum_{c=1}^C |\mathcal{R}_{\widehat{\theta}_c, \rho_c}(w) - \mathcal{R}_{\widehat{\theta}_c, 0}(w)|.$$

For each $w$, since $\widehat{\theta}_c \to \lambda(w_0, \widehat{x}_c)$ in probability, we have $\mathbb{P}(\|\widehat{\theta}_c - \lambda(w_0, \widehat{x}_c)\|_2 > 1) \to 0$. Therefore there exists compact set $\Theta_c$ for each $c$ such that $\widehat{\theta}_c$ is contained in $\Theta_c$ with probability going to 1. Choose $\overline{\rho}_c = 1$, since $\rho_c \to 0$, we have $\overline{\rho}_c \geq \rho_c$ eventually. Therefore, by Lemma C.2, for each $c$ with probability going to 1

$$|\mathcal{R}_{\widehat{\theta}_c, \rho_c}(w) - \mathcal{R}_{\widehat{\theta}_c, 0}(w)| \leq \sqrt{2\rho_c/m}\|\lambda(w, \widehat{x}_c)\|_2,$$

where the above constant $m$ can be chosen independent of $c$ due to the finite cardinality assumption of $\mathcal{X}$. Since the function $\lambda(w, \widehat{x}_c)$ is continuous in $w$ for any $\widehat{x}_c$, we have $\|\lambda(w, \widehat{x}_c)\|_2$ is bounded for all $w$ ranging over a compact set $W \subset \mathcal{W}$. Thus for each $c$ with probability going to 1, we have

$$\sup_{w \in W} |\mathcal{R}_{\widehat{\theta}_c, \rho_c}(w) - \mathcal{R}_{\widehat{\theta}_c, 0}(w)| \leq \sqrt{2\rho_c/m} \sup_{w \in W} \|\lambda(w, \widehat{x}_c)\|_2.$$

Since $\rho_c \to 0$, we have for each $c$

$$\sup_{w \in W} |\mathcal{R}_{\widehat{\theta}_c, \rho_c}(w) - \mathcal{R}_{\widehat{\theta}_c, 0}(w)| = o_{\mathbb{P}}(1).$$

Thus $\sup_{w \in W} \|t^\star - \widehat{t}\|_1 = o_{\mathbb{P}}(1)$. On the other hand, since $\sup_{w \in W} \mathcal{R}_{\widehat{\theta}_c, 0}(w)$ is $O_{\mathbb{P}}(1)$, we have $\sup_{w \in W} \|\widehat{t}\|_1 = O_{\mathbb{P}}(1)$. Therefore as $\varepsilon \to 0, \rho_c \to 0$,

$$\sup_{w \in W} |R_{\widehat{\theta}, \varepsilon, \rho}(w) - R_{\widehat{\theta}, 0, 0}(w)| = o_{\mathbb{P}}(1)$$

for any compact set $W$. Next, since $\widehat{\theta}_c \to \lambda(w_0, \widehat{x}_c)$ in probability, we have by continuous mapping theorem

$$\nabla \Psi(\widehat{\theta}_c) \to \nabla \Psi(\lambda(w_0, \widehat{x}_c)) \text{ in probability.}$$

Besides, by the strong law of large number,

$$\widehat{p}_c \to \mathbb{P}(X = \widehat{x}_c) \text{ almost surely.}$$

Recall that

$$R_{\widehat{\theta}, 0, 0}(w) = \mathbb{E}_{\widehat{\mathbb{P}}}[\ell_\lambda(X, Y, w)] = \sum_{c=1}^{C} \widehat{p}_c \mathbb{E}_{\widehat{\mathbb{P}}_{Y | \widehat{x}_c}} [\ell_\lambda(\widehat{x}_c, Y, w)]$$

$$= \sum_{c=1}^{C} \widehat{p}_c \left( \Psi(\lambda(w, \widehat{x}_c)) - \langle \nabla \Psi(\widehat{\theta}_c), \lambda(w, \widehat{x}_c) \rangle \right).$$

Therefore, for each $w$, we have

$$R_{\widehat{\theta}, 0, 0}(w) \to R(w) \text{ in probability,}$$

where

$$R(w) = \mathbb{E}_{\mathbb{P}}[\ell_\lambda(X, Y, w)] = \sum_{c=1}^{C} \mathbb{P}(X = \widehat{x}_c) \left( \Psi(\lambda(w, \widehat{x}_c)) - \langle \nabla \Psi(\lambda(w_0, \widehat{x}_c)), \lambda(w, \widehat{x}_c) \rangle \right).$$

Since for each $c$,

$$w_0 = \min_{w \in \mathcal{W}} \Psi(\lambda(w, \widehat{x}_c)) - \langle \nabla \Psi(\lambda(w_0, \widehat{x}_c)), \lambda(w, \widehat{x}_c) \rangle$$

Therefore $w_0$ solves $\min_{w \in \mathcal{W}} R(w)$. If $R(w)$ admits an unique solution, then clearly $w_0$ is such a solution. Since $R_{\widehat{\theta}, 0, 0}(\cdot)$ is convex, by [2, Theorem II.1],

$$\sup_{w \in W} |R_{\widehat{\theta}, 0, 0}(w) - R(w)| = o_{\mathbb{P}}(1)$$

for any compact set $W$. Thus by triangle inequality

$$\sup_{w \in W} |R_{\widehat{\theta}, \varepsilon, \rho}(w) - R(w)| = o_{\mathbb{P}}(1)$$

for any compact set $W$. Let $B$ denote the unit closed ball in $\mathbb{R}^q$, then $w_0 + \eta B$ is compact for any $\eta > 0$. Thus $R_{\widehat{\theta}, \varepsilon, \rho}(w) - R(w) = o_{\mathbb{P}}(1)$ uniformly over $w_0 + \eta B$. Since $R(w)$ is convex and $w_0$ is its unique optimal solution, we have

$$\inf_{w \in w_0 + \eta B \setminus \frac{\eta}{2} B} R(w) > R(w_0).$$

Therefore, with probability going to 1,

$$\inf_{w \in w_0 + \frac{\eta}{2} B} R_{\widehat{\theta}, \varepsilon, \rho}(w) < \inf_{w \in w_0 + \eta B \setminus \frac{\eta}{2} B} R_{\widehat{\theta}, \varepsilon, \rho}(w).$$

Thus by convexity of $R_{\widehat{\theta},\varepsilon,\rho}$, also

$$\inf_{w \in w_0 + \frac{\eta}{2}B} R_{\widehat{\theta},\varepsilon,\rho}(w) < \inf_{w \notin w_0 + \eta B} R_{\widehat{\theta},\varepsilon,\rho}(w).$$

Thus the solution $w^*$ that solves $\inf_{w \in \mathcal{W}} R_{\widehat{\theta},\varepsilon,\rho}(w)$ satisfies

$$\mathbb{P}(\|w^* - w_0\|_2 \leq \frac{\eta}{2}) \to 1.$$

Since $\eta$ is chosen arbitrarily, we conclude that $w^* \to w_0$ in probability. $\qquad\square$

*Proof of Lemma 4.4.* Denote

$$W_c = \sqrt{N_c}\left(\frac{\sum_{\widehat{x}_i = \widehat{x}_c} T(\widehat{y}_i)}{N_c} - \mathbb{E}_{f(\cdot\,|\theta_c)}[T(Y)]\right).$$

W.l.o.g. we can assume that $\mathbb{E}_{f(\cdot\,|\theta_c)}[T(Y)] = 0$. We first show the joint convergence

$$(W_1^\top, \ldots, W_C^\top)^\top \xrightarrow{d.} \mathcal{N}(0, G) \qquad \text{as} \qquad N \to \infty,$$

where $G$ is a block-diagonal matrix with diagonal blocks given by $G_c = \mathrm{Cov}_{f(\cdot\,|\theta_c)}(T(Y)), c = 1, \ldots, C$. Note that

$$N_c/N \to \mathbb{P}(X = \widehat{x}_c) > 0 \qquad \text{a.s. for each } c.$$

For convenience denote $r_c = \mathbb{P}(X = \widehat{x}_c)$. We let

$$\tilde{W}_c = \sqrt{\lfloor r_c N \rfloor} \cdot \frac{\sum_{\widehat{x}_i = \widehat{x}_c} T(\widehat{y}_i)}{\lfloor r_c N \rfloor} = \frac{\sum_{\widehat{x}_i = \widehat{x}_c} T(\widehat{y}_i)}{\sqrt{\lfloor r_c N \rfloor}}.$$

Let $\left[\sum_{\widehat{x}_i = \widehat{x}_c} T(\widehat{y}_i)\right]_{\lfloor r_c N \rfloor}$ be the sum of the first $\lfloor r_c N \rfloor$ samples of $T(\widehat{y}_i)$ such that $\widehat{x}_i = \widehat{x}_c$. If $N_c < \lfloor r_c N \rfloor$, we add additional $\lfloor r_c N \rfloor - N_c$ independent copies of $T(Y)$ where $Y \sim f(\cdot\,|\theta_c)$ to the sum $\sum_{\widehat{x}_i = \widehat{x}_c} T(\widehat{y}_i)$, and denote it by $\left[\sum_{\widehat{x}_i = \widehat{x}_c} T(\widehat{y}_i)\right]_{\lfloor r_c N \rfloor}$ as well. Denote

$$\bar{W}_c = \frac{\left[\sum_{\widehat{x}_i = \widehat{x}_c} T(\widehat{y}_i)\right]_{\lfloor r_c N \rfloor}}{\sqrt{\lfloor r_c N \rfloor}}.$$

Note that $\bar{W}_1, \ldots, \bar{W}_C$ are independent, by i.i.d central limit theorem

$$\left(\bar{W}_1^\top, \ldots, \bar{W}_C^\top\right)^\top \xrightarrow{d.} \mathcal{N}(0, G) \qquad \text{as} \qquad N \to \infty,$$

where $G$ is a block-diagonal matrix with $G_c = \mathrm{Cov}_{f(\cdot\,|\theta_c)}(T(Y))$. We next show that

$$\tilde{W}_c - \bar{W}_c = o_{\mathbb{P}}(1).$$

Note that

$$\tilde{W}_c - \bar{W}_c = \frac{\left[\sum_{\widehat{x}_i = \widehat{x}_c} T(\widehat{y}_i)\right]_{\lfloor r_c N \rfloor} - \sum_{\widehat{x}_i = \widehat{x}_c} T(\widehat{y}_i)}{\sqrt{\lfloor r_c N \rfloor}}.$$

By Chebyshev inequality

$$\mathbb{P}(\|\tilde{W}_c - \bar{W}_c\|_2 > \epsilon) \leq \frac{\mathbb{E}\left[\left\|\left[\sum_{\widehat{x}_i = \widehat{x}_c} T(\widehat{y}_i)\right]_{\lfloor r_c N \rfloor} - \sum_{\widehat{x}_i = \widehat{x}_c} T(\widehat{y}_i)\right\|_2^2\right]}{\epsilon^2 \lfloor r_c N \rfloor}$$

$$= \frac{\mathbb{E}\left[\mathbb{E}\left[\left\|\left[\sum_{\widehat{x}_i = \widehat{x}_c} T(\widehat{y}_i)\right]_{\lfloor r_c N \rfloor} - \sum_{\widehat{x}_i = \widehat{x}_c} T(\widehat{y}_i)\right\|_2^2\right]\Big| N_c\right]}{\epsilon^2 \lfloor r_c N \rfloor}$$

$$= \frac{\mathbb{E}[\|T(\widehat{y}_i)\|_2^2]}{\epsilon^2} \frac{\mathbb{E}[|\lfloor r_c N \rfloor - N_c|]}{\lfloor r_c N \rfloor}.$$

Since $N_c/\lfloor r_c N \rfloor \to 1$ almost surely, by dominated convergence theorem

$$\frac{\mathbb{E}[|\lfloor r_c N \rfloor - N_c|]}{\lfloor r_c N \rfloor} \to 0.$$

Thus
$$\mathbb{P}(\|\tilde{W}_c - \bar{W}_c\|_2 > \epsilon) \to 0 \qquad \text{as} \qquad N \to \infty,$$
which means that $\tilde{W}_c - \bar{W}_c = o_\mathbb{P}(1)$. Thus by Slutsky's lemma
$$\left(\tilde{W}_1^\top, \ldots, \tilde{W}_C^\top\right)^\top \xrightarrow{d.} \mathcal{N}(0, G) \qquad \text{as} \qquad N \to \infty.$$

Finally, since $W_c = (1 + o_\mathbb{P}(1))\tilde{W}_c$, by Slutsky's lemma,
$$\left(W_1^\top, \ldots, W_C^\top\right)^\top \xrightarrow{d.} \mathcal{N}(0, G) \qquad \text{as} \qquad N \to \infty.$$

Now note that
$$\widehat{\theta}_c = (\nabla\Psi)^{-1}\left((N_c)^{-1}\sum_{\widehat{x}_i = \widehat{x}_c} T(\widehat{y}_i)\right)$$
and
$$\theta_c = (\nabla\Psi)^{-1}\left(\mathbb{E}_{f(\cdot|\theta_c)}[T(Y)]\right).$$
Also note that the vector-valued function $(\nabla\Psi)^{-1}(\cdot)$ is continuously differentiable at $\mathbb{E}_{f(\cdot|\theta_c)}[T(Y)]$, therefore, by the delta method
$$\left(\sqrt{N_1}(\widehat{\theta}_1 - \theta_1)^\top, \ldots, \sqrt{N_C}(\widehat{\theta}_C - \theta_C)^\top\right)^\top \xrightarrow{d.} D \cdot \mathcal{N}(0, G),$$
where $D$ is a block-diagonal matrix with diagonal elements given by
$$D_c = J(\nabla\Psi)^{-1}(\mathbb{E}_{f(\cdot|\theta_c)}[T(Y)])$$
the Jacobian matrix of $(\nabla\Psi)^{-1}$ evaluated at $\mathbb{E}_{f(\cdot|\theta_c)}[T(Y)]$. Thus
$$V_c = D_c \text{Cov}_{f(\cdot|\theta_c)}(T(Y))D_c^\top.$$
Note that by Lemma B.3, we find
$$\text{KL}(f(\cdot|\theta_c) \,\|\, f(\cdot|\widehat{\theta}_c)) = \langle \theta_c - \widehat{\theta}_c, \mu_c\rangle + \Psi(\widehat{\theta}_c) - \Psi(\theta_c).$$
Note that $\Psi$ is infinitely-many differentiable, we have the follow Taylor expansion
$$\Psi(\widehat{\theta}_c) - \Psi(\theta_c) = \langle \widehat{\theta}_c - \theta_c, \mu_c\rangle + \frac{1}{2}\langle \widehat{\theta}_c - \theta_c, \nabla^2\Psi(\theta_c + \eta(\widehat{\theta}_c - \theta_c))(\widehat{\theta}_c - \theta_c)\rangle,$$
where $\eta$ is a random variable with values between 0 and 1. Therefore
$$\text{KL}(f(\cdot|\theta_c) \,\|\, f(\cdot|\widehat{\theta}_c)) = \frac{1}{2}\langle \widehat{\theta}_c - \theta_c, \nabla^2\Psi(\theta_c + \eta(\widehat{\theta}_c - \theta_c))(\widehat{\theta}_c - \theta_c)\rangle.$$
Because $\sqrt{N_c}(\widehat{\theta}_c - \theta_c) \xrightarrow{d.} \mathcal{N}(0, V_c)$, and $\nabla^2\Psi(\cdot)$ is continuous, we have
$$\nabla^2\Psi(\theta_c + \eta(\widehat{\theta}_c - \theta_c)) = \nabla^2\Psi(\theta_c) + o_\mathbb{P}(1).$$
Moreover, since we have the joint convergence
$$\left(\sqrt{N_1}(\widehat{\theta}_1 - \theta_1)^\top, \ldots, \sqrt{N_C}(\widehat{\theta}_C - \theta_C)^\top\right)^\top \xrightarrow{d.} \mathcal{N}(0, V),$$
by continuous mapping theorem
$$\left(N_1 \times \text{KL}(f(\cdot|\theta_1) \,\|\, f(\cdot|\widehat{\theta}_1)), \ldots, N_C \times \text{KL}(f(\cdot|\theta_C) \,\|\, f(\cdot|\widehat{\theta}_C))\right)^\top \xrightarrow{d.} Z \quad \text{as} \quad N \to \infty,$$
where $Z = (Z_1, \ldots, Z_C)^\top$ with $Z_c = \frac{1}{2}R_c^\top\nabla^2\Psi(\theta_c)R_c$, $R_c \sim \mathcal{N}(0, V_c)$ and are independent for $c = 1, \ldots, C$. $\qquad\square$

Before proving the result on the worst-case distribution in Theorem 4.5, we first prove the worst-case conditional measure that maximize problem (9).

**Proposition C.3** (Worst-case conditional distribution)**.** For any $w \in \mathcal{W}$ and $\rho_c \in \mathbb{R}_{++}$, then the supremum problem (9) is attained by $\mathbb{Q}^\star_{Y|\widehat{x}_c} \sim f(\,\cdot\,|\theta^\star_c)$ with $\theta^\star_c = \widehat{\theta}_c - \lambda(w, \widehat{x}_c)/\gamma^\star_c$, where $\gamma^\star_c > 0$ is the solution of the nonlinear algebraic equation

$$\Psi\big(\widehat{\theta}_c - \gamma^{-1}\lambda(w, \widehat{x}_c)\big) + \gamma^{-1}\big\langle \nabla\Psi\big(\widehat{\theta}_c - \gamma^{-1}\lambda(w, \widehat{x}_c)\big), \lambda(w, \widehat{x}_c)\big\rangle = \Psi(\widehat{\theta}_c) - \rho_c. \qquad (A.9)$$

*Proof of Proposition C.3.* Reminding that problem (9) is written as

$$\sup_{\mathbb{Q}_{Y|\widehat{x}_c} \in \mathbb{B}_{Y|\widehat{x}_c}} \mathbb{E}_{\mathbb{Q}_{Y|\widehat{x}_c}}\left[\ell_\lambda(\widehat{x}_c, Y, w)\right].$$

In the first step, we show that $\mathbb{Q}^\star_{Y|\widehat{x}_c}$ is feasible in problem (9), which means that $\mathbb{Q}^\star_{Y|\widehat{x}_c} \in \mathbb{B}_{Y|\widehat{x}_c}$. Indeed, we find that

$$\mathrm{KL}(\mathbb{Q}^\star_{Y|\widehat{x}_c} \,\|\, \widehat{\mathbb{P}}_{Y|\widehat{x}_c}) = -\Psi\Big(\widehat{\theta}_c - \frac{\lambda(w, \widehat{x}_c)}{\gamma^\star_c}\Big) - \frac{1}{\gamma^\star_c}\Big\langle \nabla\Psi\Big(\widehat{\theta}_c - \frac{\lambda(w, \widehat{x}_c)}{\gamma^\star_c}\Big), \lambda(w, \widehat{x}_c)\Big\rangle + \Psi(\widehat{\theta}_c) = \rho_c,$$

where the first equality exploits the expression of the KL divergence between two distributions from the same family in Lemma B.3, and the second equality follows from the fact that $\gamma^\star_c$ solves (A.9).

Proposition 3.2 asserts that the worst-case conditional expected log-loss problem (9) is equivalent to the convex program (10). Noticing that (A.9) is the first-order optimality condition of problem (10), thus, by definition, $\gamma^\star_c$ is the minimizer of (10). The objective value of $\mathbb{Q}^\star_{Y|\widehat{x}_c}$ in (9) amounts to

$$\begin{aligned}
\mathbb{E}_{\mathbb{Q}^\star_{Y|\widehat{x}_c}}\left[\ell_\lambda(\widehat{x}_c, Y, w)\right] &= \Psi(\lambda(w, \widehat{x}_c)) - \big\langle \mathbb{E}_{\mathbb{Q}^\star_{Y|\widehat{x}_c}}[T(Y)], \lambda(w, \widehat{x}_c)\big\rangle \\
&= \Psi(\lambda(w, \widehat{x}_c)) - \Big\langle \nabla\Psi\Big(\widehat{\theta}_c - \frac{\lambda(w, \widehat{x}_c)}{\gamma^\star_c}\Big), \lambda(w, \widehat{x}_c)\Big\rangle \\
&= \gamma^\star_c\big(\rho_c - \Psi(\widehat{\theta}_c)\big) + \gamma^\star_c\Psi\Big(\widehat{\theta}_c - \frac{\lambda(w, \widehat{x}_c)}{\gamma^\star_c}\Big) + \Psi(\lambda(w, \widehat{x}_c)),
\end{aligned}$$

where the first equality follows by substituting the expression of $\ell_\lambda$ and the linearity of the expectation operator, the second equality follows from the convex conjugate relationship between the expectation parameters and the log-partition function $\Psi$, and the last equality follows from the fact that $\gamma^\star_c$ solves (A.9). Notice that the last expression coincide with the objective value of (10) evaluated at the optimal solution $\gamma^\star_c$. This observation implies that $\mathbb{Q}^\star_{Y|\widehat{x}_c}$ attains the optimal value in (9). $\qquad\square$

Next, we establish the following result on the optimal solution of the support function $h_\mathcal{Q}$ of the set $\mathcal{Q}$ defined as in Lemma B.1.

**Lemma C.4** (Support point of $\mathcal{Q}$)**.** Let $\mathcal{Q}$ be defined as in (A.1). For any $t \in \mathbb{R}^C$, if there exist $\alpha^\star \in \mathbb{R}$ and $\beta^\star \in \mathbb{R}_{++}$ that solve the following system of nonlinear algebraic equation

$$\sum_{c=1}^C \widehat{p}_c \exp\Big(\frac{t_c - \alpha}{\beta} - \rho_c - 1\Big) - 1 = 0 \qquad (A.10a)$$

$$\sum_{c=1}^C \widehat{p}_c(t_c - \alpha) \exp\Big(\frac{t_c - \alpha}{\beta} - \rho_c - 1\Big) - (\varepsilon + 1)\beta = 0 \qquad (A.10b)$$

then the optimal solution $q^\star \in \mathcal{Q}$ that attains $t^\top q^\star = h_\mathcal{Q}(t)$ is

$$q^\star_c = \widehat{p}_c \exp\Big(\frac{t_c - \alpha^\star}{\beta^\star} - \rho_c - 1\Big) \qquad \forall c = 1, \dots, C. \qquad (A.10c)$$

*Proof of Lemma C.4.* By definition of $q^\star$ in (A.10c), one can verify that $q^\star \geq 0$ and that $\sum_{c=1}^C q^\star_c = 1$, where the equality follows from (A.10a). Moreover,

$$\begin{aligned}
\sum_{c=1}^C q^\star_c(\log q^\star_c - \log\widehat{p}_c + \rho_c) &= \sum_{c=1}^C \widehat{p}_c\Big(\frac{t_c - \alpha^\star}{\beta^\star} - 1\Big)\exp\Big(\frac{t_c - \alpha^\star}{\beta^\star} - \rho_c - 1\Big) \\
&= \sum_{c=1}^C \widehat{p}_c\Big(\frac{t_c - \alpha^\star}{\beta^\star}\Big)\exp\Big(\frac{t_c - \alpha^\star}{\beta^\star} - \rho_c - 1\Big) - 1 = \varepsilon,
\end{aligned}$$

where the equalities follow from the definition of $q^\star$ in (A.10c), and the equations (A.10a) and (A.10b), respectively. This implies that $q^\star \in \mathcal{Q}$.

It now remains to show that $t^\top q^\star = h_\mathcal{Q}(t)$. By Lemma B.1, we have

$$
h_\mathcal{Q}(t) = \begin{cases} \inf & \alpha + \varepsilon\beta + \beta \sum_{c=1}^{C} \widehat{p}_c \exp\left(\frac{t_c - \alpha}{\beta} - \rho_c - 1\right) \\ \text{s.t.} & \alpha \in \mathbb{R}, \ \beta \in \mathbb{R}_{++}. \end{cases}
$$

If $(\alpha^\star, \beta^\star) \in \mathbb{R} \times \mathbb{R}_{++}$ is the solution of (A.10a)-(A.10b), then $(\alpha^\star, \beta^\star)$ satisfy the Karush-Kuhn-Tucker condition of the above infimum optimization problem, and thus we have

$$
h_\mathcal{Q}(t) = \alpha^\star + \varepsilon\beta^\star + \beta^\star \sum_{c=1}^{C} \widehat{p}_c \exp\left(\frac{t_c - \alpha^\star}{\beta^\star} - \rho_c - 1\right).
$$

Moreover, we find

$$
\begin{aligned}
\sum_{c=1}^{C} t_c q_c^\star &= \sum_{c=1}^{C} t_c \widehat{p}_c \exp\left(\frac{t_c - \alpha^\star}{\beta^\star} - \rho_c - 1\right) \\
&= (\varepsilon + 1)\beta^\star + \alpha^\star \sum_{c=1}^{C} \widehat{p}_c \exp\left(\frac{t_c - \alpha^\star}{\beta^\star} - \rho_c - 1\right) \\
&= \alpha^\star + \varepsilon\beta^\star + \beta^\star \sum_{c=1}^{C} \widehat{p}_c \exp\left(\frac{t_c - \alpha^\star}{\beta^\star} - \rho_c - 1\right) = h_\mathcal{Q}(t),
\end{aligned}
$$

where the first equality follows from the definition of $q^\star$, the second equality follows from (A.10b) and the third equality follows from (A.10a). This observation completes the proof. $\qquad\square$

*Proof of Theorem 4.5.* It is easy to verify that $\mathbb{Q}^\star$ is a probability measure because each $\delta_{\widehat{x}_c}$ and $\mathbb{Q}^\star_{Y|\widehat{x}_c}$ is a probability measure, and $\sum_{c=1}^{C} \widehat{p}_c \exp\left((t_c^\star - \alpha^\star)/\beta^\star - \rho_c - 1\right) = 1$ since $\alpha^\star, \beta^\star$ solves

$$
\sum_{c=1}^{C} \widehat{p}_c \exp\left(\beta^{-1}(t_c^\star - \alpha) - \rho_c - 1\right) - 1 = 0 \tag{A.11}
$$

$$
\sum_{c=1}^{C} \widehat{p}_c(t_c^\star - \alpha) \exp\left(\beta^{-1}(t_c^\star - \alpha) - \rho_c - 1\right) - (\varepsilon + 1)\beta = 0, \tag{A.12}
$$

If we set $\mathbb{Q}_X^\star = \sum_{c=1}^{C} \widehat{p}_c \exp\left((t_c^\star - \alpha^\star)/\beta^\star - \rho_c - 1\right)\delta_{\widehat{x}_c}$, then we have

$$
\mathbb{Q}^\star(\{\widehat{x}_c\} \times A) = \mathbb{Q}_X^\star(\{\widehat{x}_c\})\mathbb{Q}^\star_{Y|\widehat{x}_c}(A) \quad \forall A \in \mathcal{F}(\mathcal{Y}), \ \forall c.
$$

Moreover, because $\mathbb{Q}^\star_{Y|\widehat{x}_c}$ is constructed using Proposition C.3, we have $\mathrm{KL}(\mathbb{Q}_{Y|\widehat{x}_c} \| \widehat{\mathbb{P}}_{Y|\widehat{x}_c}) \leq \rho_c$ for all $c$. Furthermore, we also have

$$
\begin{aligned}
\mathrm{KL}(\mathbb{Q}_X^\star \| \widehat{\mathbb{P}}_X) + \mathbb{E}_{\mathbb{Q}_X^\star}\left[\sum_{c=1}^{C} \rho_c \mathbb{1}_{\widehat{x}_c}(X)\right] &= \sum_{c=1}^{C} \widehat{p}_c\left(\frac{t_c^\star - \alpha^\star}{\beta^\star} - 1\right)\exp\left(\frac{t_c^\star - \alpha^\star}{\beta^\star} - \rho_c - 1\right) \\
&= \sum_{c=1}^{C} \widehat{p}_c\left(\frac{t_c^\star - \alpha^\star}{\beta^\star}\right)\exp\left(\frac{t_c^\star - \alpha^\star}{\beta^\star} - \rho_c - 1\right) - 1 = \varepsilon,
\end{aligned}
$$

where the equalities follow from the construction of $\mathbb{Q}_X^\star$ and the equations (A.11) and (A.12), respectively. This implies that $\mathbb{Q}^\star \in \mathbb{B}(\widehat{\mathbb{P}})$.

It now remains to show that $\mathbb{Q}^\star$ is optimal. For any weight $w$, by the definition of $t_c^\star$, we have

$$
t_c^\star = \mathbb{E}_{\mathbb{Q}^\star_{Y|\widehat{x}_c}}\left[\ell_\lambda(\widehat{x}_c, Y, w)\right] = \sup_{\mathbb{Q}_{Y|\widehat{x}_c} \in \mathbb{B}_{Y|\widehat{x}_c}} \mathbb{E}_{\mathbb{Q}_{Y|\widehat{x}_c}}\left[\ell_\lambda(\widehat{x}_c, Y, w)\right]
$$

We thus find

$$\max_{\mathbb{Q}\in\mathbb{B}(\widehat{\mathbb{P}})} \mathbb{E}_{\mathbb{Q}}\Big[\ell_\lambda(X,Y,w)\Big] = \sup_{\mathbb{Q}_X\in\mathbb{B}_X} \mathbb{E}_{\mathbb{Q}_X}\left[\sup_{\mathbb{Q}_{Y|X}\in\mathbb{B}_{Y|X}} \mathbb{E}_{\mathbb{Q}_{Y|X}}\left[\ell_\lambda(X,Y,w)\right]\right]$$

$$= \sup_{\mathbb{Q}_X\in\mathbb{B}_X} \mathbb{E}_{\mathbb{Q}_X}\left[\sum_{c=1}^{C} t_c^\star \mathbb{1}_{\widehat{x}_c}(X)\right]$$

$$= \sup_{q\in\mathcal{Q}} q^\top t^\star \tag{A.13}$$

$$= \sum_{c=1}^{C} \widehat{p}_c t_c^\star \exp\left(\frac{t_c^\star - \alpha^\star}{\beta^\star} - \rho_c - 1\right) \tag{A.14}$$

$$= \mathbb{E}_{\mathbb{Q}_X^\star}\left[\sum_{c=1}^{C} t_c^\star \mathbb{1}_{\widehat{x}_c}(X)\right] \tag{A.15}$$

$$= \mathbb{E}_{\mathbb{Q}_X^\star}\Big[\mathbb{E}_{\mathbb{Q}_{Y|X}^\star}\left[\ell_\lambda(X,Y,w)\right]\Big] = \mathbb{E}_{\mathbb{Q}^\star}\Big[\ell_\lambda(X,Y,w)\Big].$$

where the set $\mathcal{Q}$ in (A.13) is defined as in (A.1). Equality (A.14) follows from Lemma C.4 and from the definition of $\alpha^\star$ and $\beta^\star$ that solve (A.11)-(A.12). Equality (A.15) follows from the definition of $\mathbb{Q}_X^\star$. The proof is completed. $\qquad\square$

# D  Auxiliary Results

**Lemma D.1** (Locally strongly convex parameter). If $\Psi$ is locally strongly smooth, and at $\widehat{\theta}$, the smoothness parameter is $\sigma$, then $\phi$ is locally strongly convex at $\widehat{\mu} = \nabla\Psi(\widehat{\theta})$ with strongly convex parameter $1/\sigma$ in a sufficiently small neighbourhood of $\widehat{\mu}$.

*Proof of Lemma D.1.* The proof follows directly from the proof of [9, Theorem 4.1]. By the definition of locally strongly smooth, for some $\Theta' \subseteq \Theta$ neighborhood of $\widehat{\theta}$, we have for $\theta \in \Theta'$

$$\Psi(\theta) \le \Psi(\widehat{\theta}) + \big\langle\nabla\Psi(\widehat{\theta}), \theta - \widehat{\theta}\big\rangle + \frac{\sigma}{2}\|\theta - \widehat{\theta}\|_2^2.$$

Since $\widehat{\mu} = \nabla\Psi(\widehat{\theta})$ and $\phi(\widehat{\mu}) = \langle\widehat{\mu}, \widehat{\theta}\rangle - \Psi(\widehat{\theta})$, we have

$$\phi(\mu) = \sup_{\theta\in\Theta} \big(\langle\mu,\theta\rangle - \Psi(\theta)\big)$$

$$\ge \sup_{\theta\in\Theta'} \Big(\langle\mu,\theta\rangle - \Psi(\widehat{\theta}) - \langle\widehat{\mu}, \theta - \widehat{\theta}\rangle - \frac{\sigma}{2}\|\theta - \widehat{\theta}\|_2^2\Big)$$

$$= \langle\widehat{\mu}, \widehat{\theta}\rangle - \Psi(\widehat{\theta}) + \sup_{\theta\in\Theta'} \Big(\langle\mu,\theta\rangle - \langle\widehat{\mu}, \theta\rangle - \frac{\sigma}{2}\|\theta - \widehat{\theta}\|_2^2\Big)$$

$$= \phi(\widehat{\mu}) + \langle\widehat{\theta}, \mu - \widehat{\mu}\rangle + \sup_{\theta\in\Theta'} \Big(\langle\mu - \widehat{\mu}, \theta - \widehat{\theta}\rangle - \frac{\sigma}{2}\|\theta - \widehat{\theta}\|_2^2\Big).$$

In the last step, note that $\widehat{\theta} = \nabla\phi(\widehat{\mu})$. Taking $\theta - \widehat{\theta} = \alpha(\mu - \widehat{\mu})$ where $\alpha = 1/\sigma$. $\theta \in \Theta'$ if $\mu - \widehat{\mu}$ is sufficiently small. We have

$$\sup_{\theta\in\Theta'} \Big(\langle\mu - \widehat{\mu}, \theta - \widehat{\theta}\rangle - \frac{\sigma}{2}\|\theta - \widehat{\theta}\|_2^2\Big) \ge (\alpha - \frac{\sigma}{2}\alpha^2)\|\mu - \widehat{\mu}\|_2^2 = \frac{1}{2\sigma}\|\mu - \widehat{\mu}\|_2^2.$$

Therefore $\phi$ is locally strongly convex at $\widehat{\mu}$ with strongly convex parameter $1/\sigma$. $\qquad\square$

In Proposition 4.2, since $\Psi$ is locally Lipschitz continuous, we have that $\Psi$ is locally strongly smooth with smoothness parameter $\sigma_c$ at $\widehat{\theta}_c$, where $\sigma_c$ can be chosen as the local Lipschitz constant for a neighborhood around $\widehat{\theta}_c$. By Lemma D.1 and the proof of Proposition 4.2, for sufficiently small $\rho_c, c = 1, \ldots, C$, we can choose $m$ explicitly as $m = \min_c 1/\sigma_c$, thus $\kappa_2 = \sqrt{2\max_c \rho_c \cdot \max_c \sigma_c}$.