[Reviews · NeurIPS 2020]

Review 1

Summary and Contributions: The paper proposes a framework to incorporate parametric information in DRO for supervised machine-learning. DRO is a promising alternative to the traditional ERM, as it typically enjoys better performance under small-sample regimes, or out of distribution corruption. This paper considers the parametric case of exponential-families and proposes to use ambiguity sets which respect this parametric structure.

Strengths: The paper is well-written and easy to follow.

Weaknesses: - The contribution of the paper is too marginal. DRO with KL ambiguity sets has been extensively studied. For example, the nonparametric case, see Faury et al "Distributionally Robust Counterfactual Risk Minimization" (AAAI 2020) and Si et al "Distributionally Robust Policy Evaluation and Learning in Offline Contextual Bandits" (ICML 2020). These papers propose explicit wort-case distributions, tractable algorithms for the modified robust ERM (via reweighting), and complete theoretical analysis thereof (consistency, sample complexity, etc.). - Moreover, in the particular parametric case of DRO on exponential families (the subject of this manuscript) has already been studied in Hu et al. "Kullback-Leibler Divergence Constrained Distributionally Robust Optimization" - The paper lacks solid theoretical background. Since everything is parametric, I'd expect explicit rates of convergence involvind all probalem complexity parameters (n, m, p, etc.) To make the rest of my points clear, let me recall the following notations are used in the paper: - n: the dimensionality of the covariate (i.e feature vector) X. Thus X is random vector in R^n. BTW, in the context of ML or stats, I'd use another notation here, as n conventionally stands for "sample size". - m: the dimensionality of the output variable Y. Thus X is a random vector in R^m. - p: the dimensionality of the ambient space in which the model parameter theta lives. - N: the number of samples in the training dataset - (x_1,y_1),...,(x_N,y_N): and iid sample of size N from the unknown joint distribution of X and Y. - C the number of distinct feature vectors x_i in the sample. Thus 1 <= C <= N. WLOG, tet the distinct feature vectors be x_1,...,x_C. - N_c (with 1 <= c <= C): #{i | x_i = x_c}, i.e number of examples whose features vector equals x_c. - Line 95 to 98: Since the covariates (i.e features) are continuous, we are certain to have C = N and N_c = 1 for all c. - Line 99 to 102: The model in (5) has C + dim(Theta)^C, which is rougly N + dim(Theta)^N parameters in case of continuous covariates (see previous comment). This cannot possibly work as soon as dim(Theta) > 1. - Proposition 4.2: This should be rewritten to clearly outline the dependence on the sample size N. Also, the third term on the right is mysterious. Also, at what rates do kappa_1 and kappa_2 go to zero (if they do...). - Line 227: I don't see how you let "N_c tend to infinity" in view of my above comment on N_c = 1 almost certainly (see my previous comments). The rest of the analysis (Lemma 4.4) is therefore awkward. - Why are these experimental setups presented in section 5 relevant to the subject ? - Documentation on the datasets in Table 2 should be provided (n = ?, m = ?, etc.).

Correctness: Lemma 4.4 sounds ackward. How can N_c tend to infinity when the covariates are continuous ? (See my previous comments).

Clarity: Yes.

Relation to Prior Work: No. DRO with KL ambiguity sets has been extensively studied. For example, the nonparametric case, see Faury et al "Distributionally Robust Counterfactual Risk Minimization" (AAAI 2020) and Si et al "Distributionally Robust Policy Evaluation and Learning in Offline Contextual Bandits" (ICML 2020). These papers propose explicit wort-case distributions, tractable algorithms for the modified robust ERM (via reweighting), and complete theoretical analysis thereof (consistency, sample complexity, etc.). - Moreover, in the particular parametric case of DRO on exponential families (the subject of this manuscript) has already been studied in Hu et al. "Kullback-Leibler Divergence Constrained Distributionally Robust Optimization"

Reproducibility: Yes

Additional Feedback:


Review 2

Summary and Contributions: This paper studies the robust maximum likelihood estimator for exponential families using KL ambiguity set. Comprehensive theoretical results are presented, including the tractable reformulation and statistical guarantees.

Strengths: This work presents a strong theoretical contribution to the maximum likelihood estimator in DRO setting. Both theoretical results and empirical evaluation on synthetic/real data are promising. This work is a good attempt in applying DRO techniques to classical statistical problems and could serve as a good complement in the current literature of robust maximum likelihood estimation.

Weaknesses: I think the major limitation is that the model is still restrictive in the following sense: the ambiguity set does not change the support of X, which is the same as training data, this indeed leads to the tractable reformulation, but it doesn’t consider possible contamination in the X variable. In other words, the restriction to discrete X might be inappropriate in certain applications.

Correctness: Correct.

Clarity: This paper is well written. The organization is very clear and easy to follow.

Relation to Prior Work: Satisfactory.

Reproducibility: Yes

Additional Feedback: The ambiguity set defined in (7) is good in terms of the tractability and a fully parametric structure, but is also restricted and heavily rely on the empirical samples. If the underlying distribution is continuous and the training size is small, than the ambiguity set defined in (7) might not be a good set in terms of its similarity with the true underlying distribution. Therefore, I think it would be more interesting if the authors could add a few discussion about possible extension of this DRO MLE framework for other uncertainty sets (such as the Wasserstein ambiguity set that does not restrict the support). Moreover, Huber has a lot of seminal work on robust estimation of parameters under distribution contamination, which could be highly related to the content of this paper, but are not sufficiently mentioned or cited in this paper. A minor comment: in the Poisson counting model example 3.4, the pmf actually corresponds to Poi( e^{w0^T x}), not w0^T x as written in the paper.


Review 3

Summary and Contributions: This paper considers a parametric distributionally robust optimization under the Kullback-Leibler divergence. Unlike the existing nonparametric framework, it is assumed that the considered distributions belong to the exponential family whose parameter is determined by a function of a covariate, and the nominal distribution is also a parametric distribution whose parameters are estimated from data. Parallel to the existing non-parametric results, dual reformulation, consistency and worst-case distributions are derived. The paper also discuss its connection to the exponential reweighting and variance regularization and conducts experiments for Poisson and logistic regression using UCI datasets.

Strengths: (1) The paper complements the existing non-parametric distributionally robust optimization by considering the exponential family with covariates. (2) The theoretical results build off of techniques from distributionally robust optimization under Kullback-Leibler divergence and the derivation demonstrates the authors' master on these techniques. (3) Numerical results demonstrate some superiority of the proposed approach.

Weaknesses: (1) One key issue with the current framework is the specification of the radii rho_c. Lemma 4.4 needs significant improvement. By the nature of MLE esimators, for Examples 2.2 and 2.3, the conditions should be a joint convergence for all c, but the current form is separate among all c. In addition, I would suggest providing an explicit form for Z as it is crucial to determining the radii. (2) Several issues regarding the numerical experiments are in order. First, how does the parametric framework comparing to existing non-parametric distributionally robust optimization under Kullback-Leibler divergence? Second, in the current setup, it is assumed that rho_c only depends on the sample size N_c, while my intuition is that the radii rho_c should depend on the magnitude of x_c as well, which can be inferred from the asymptotic convergence. I would suggest running the experiments with a more carefully designed tunning scheme. I will raise my score if the authors can successfully address these issues. === Edit after rebuttal ==== The rebuttal briefly addresses my concerns about the joint asymptotic convergence results and I wish to see a full exhibition in the final version. On the other hand, I am not convinced that choosing \rho_c based on x_c is unrealistic. One may try to study the asymptotics at the optimal w, in the spirit similar to [13]. Hence, I will keep my score.

Correctness: Yes to the best of my knowledge.

Clarity: Yes.

Relation to Prior Work: Yes.

Reproducibility: Yes

Additional Feedback:


Review 4

Summary and Contributions: The paper presents a regularization scheme for parameter estimation using maximum likelihood estimation (MLE). The new scheme leverages the distributionally robust optimization (DRO) [arXiv:1908.05659] framework, which is related to adversarial learning. The goal is to achieve better generalization for MLE as compared existing techniques such as L1 and L2 regularization. The paper casts the MLE learning problem within a min-max setup where the loss function is minimized with respect to an expectation (that is maximized) over an adverse distribution (element of a parametric ambiguity set). The learning scheme is applied to Poisson and logistic regression, which are trained using off-the-shelf optimization software and empirically evaluated on synthetic and UCI datasets.

Strengths: The paper casts regularized MLE within a min-max (adversarial) setting. Theoretical analysis are provided.

Weaknesses: For Poisson regression, the technique works for extremely small sample sizes (50,100) as demonstrated in Table 1, but at 500 (still very small), we see the performance difference between L2 and DRO is much less. This technique may find application in extremely small data size regimes, but it is difficult to imagine it applicable (given the higher complexity setup) to the big data regime. For logistic regression on UCI datasets (Table 2), the performance improvements are not significant.

Correctness: They appear correct to me.

Clarity: Yes.

Relation to Prior Work: Yes.

Reproducibility: Yes

Additional Feedback: Are there relations with your approach to the idea of over-parameterization? ----- update after authors' response ----- I have read through the authors' feedback and reviews. My review remains unchanged.

[Author Response · NeurIPS 2020]

We would like to thank all referees for their appreciation of our results and the useful feedback. Below is our reply.

**Reviewer 1:** Thank you for pointing us to the relevant references. Si et al. was made available only in July 2020 during ICML, while we cited Hu et al. as Reference [17] in our paper. Our setting (including the loss function, the construction of the nominal distribution and the ambiguity set) is clearly different from Faury et al. and Si et al. Results from Section 5.3 of Hu et al. focus solely on loss functions which are linear in $Y$. On the contrary, our loss function is linear in $T(Y)$, where the sufficient statistics $T$ can be a *non*linear map. Examples of exponential family of distributions with nonlinear $T$ are Gaussian, Gamma and Beta distributions. The results from Hu et al. thus are not directly applicable to robustify MLE problems. Our proof is also novel: it invokes duality results on the expectation parameter space.

To our best knowledge, Faury et al. and Si et al. provide convergence results for the *objective value*, not for the *solution*. We, however, focus on the convergence of our *solution*, and Theorem 4.3 shows that our solution is consistent. Results that provide the rate of convergence for DRO solution is very scarce in the current literature (we are only aware of (arXiv:1906.01614)) because it is significantly harder to prove. This extension goes beyond the scope of this paper.

In many practical applications, the covariate space $\mathcal{X}$ is categorical/ordinal, thus $N_c$ can be greater than 1. Moreover, model (5) needs only $(p+1)C$ parameters, not $C + p^C$ parameters as the reviewer thought ($p$ is the dimension of $\Theta$). We agree that the dependence on $N$ should be made explicit in Proposition 4.2. We update Table 2 below with values of $n$, $N$ and $m$. We will change the covariate space from $\mathbb{R}^n$ to $\mathbb{R}^d$ as you kindly recommended. Thank you!

Section 5 demonstrates the generality and power of our proposed approach. Under study are two popular applications of model estimation with exponential family of distributions: Poisson regression (see Example 3.4) is when $Y|X$ follows a Poisson distribution, and logistic regression (see Example 3.5) is when $Y|X$ follows a Bernoulli distribution.

**Reviewer 2:** We wholeheartedly agree with you that it is interesting to handle the contamination in the $X$ space. We tried it, and we encountered two technical difficulties: first, the log-partition function $\Psi$ is convex; second, there are multiplicative terms between $X$ and $Y$. Maximizing over the $X$ space to find the worst-case covariate is thus difficult. Even though this is a useful extension, providing sound and rigorous mathematical treatment requires significant breakthroughs in non-convex programming; or we need to approach this problem from a different perspective. One can think of perturbing each $\widehat{x}_c$ in a finite set but that would lead to trivial (and uninteresting) modifications of the constraints. We will discuss about the covariate contamination in the concluding remarks.

We will add relevant references from the robust statistics literature into the paper. Thank you for your suggestion! About the Poisson pmf: we use the *natural* parameter, while you are using the *expectation* parameter to characterize the Poisson distribution. Both are actually equivalent reparametrization. We will clarify this notation in the revised version.

**Reviewer 3:** 1) Thank you for the suggestion about the joint convergence. For brevity, we present here only the case when the nominal distribution is set using Example 2.2. Complete characterization of $Z$ is also provided.

**Lemma** (Asymptotic convergence). *Suppose that $|\mathcal{X}| = C$ with $\mathbb{P}(X = \widehat{x}_c) > 0$. Let $\theta_c = \lambda(w_0, \widehat{x}_c)$ and $\widehat{\mathbb{P}}$ be defined as in Example 2.2. Let $V_c = D_c \mathrm{Cov}_{f(\cdot|\theta_c)}(T(Y))D_c^\top$, where $D_c = J(\nabla \Psi)^{-1}(\mathbb{E}_{f(\cdot|\theta_c)}[T(Y)])$ and $J$ denotes the Jacobian operator. Then the following joint convergence holds*
$$\left(N_1 \times \mathrm{KL}(f(\cdot|\theta_1) \parallel f(\cdot|\widehat{\theta}_1)), \ldots, N_C \times \mathrm{KL}(f(\cdot|\theta_C) \parallel f(\cdot|\widehat{\theta}_C))\right)^\top \xrightarrow{d.} Z \qquad as \qquad N \to \infty,$$
*where $Z = (Z_1, \ldots, Z_C)^\top$ with $Z_c = \frac{1}{2}R_c^\top \nabla^2 \Psi(\theta_c)R_c$, $R_c$ are independent and $R_c \sim \mathcal{N}(0, V_c)$.*

2) By Proposition 4.1, nonparametric KL is equivalent to setting $\rho_c = 0$ and tune only with $\varepsilon > 0$. We rerun the experiment for logistic regression and update Table 2 as below: the column 'KL' reports the performance of nonparametric KL with $\varepsilon \in [10^{-4}, 10]$ with 10 logarithmic scale points. Intuitively, $\rho_c$ should depend on $\widehat{x}_c$, but the dependence is dictated by the unknown value $w_0$. Choosing $\rho_c$ based on $\widehat{x}_c$ is thus unrealistic.

**Reviewer 4:** Regarding the weakness: overfitting is a severe problem when the number of i.i.d. samples is relatively small compared to the dimension. This weakness (the improvement decreases as $N$ gets large) is thus pertinent to all methods aiming to combat overfitting, including regularization and DRO, and not just to our proposed method.

| | AUC | | | | | CCR | | | | |
|---|---|---|---|---|---|---|---|---|---|---|
| Dataset | DRO | KL | $L_1$ | $L_2$ | MLE | DRO | KL | $L_1$ | $L_2$ | MLE |
| australian ($N = 690, n = 14$) | **92.74** | 92.62 | 92.73 | 92.71 | 92.61 | **85.75** | 85.72 | 85.52 | 85.60 | 85.72 |
| banknote ($N = 1372, n = 4$) | **98.46** | **98.46** | 98.43 | 98.45 | 98.45 | 94.31 | 94.32 | 94.16 | **94.35** | 94.32 |
| climate ($N = 540, n = 18$) | 94.30 | 82.77 | **94.85** | 94.13 | 82.76 | **95.04** | 93.89 | 94.85 | 94.83 | 93.89 |
| german ($N = 1000, n = 19$) | **75.75** | 75.68 | 75.74 | 75.74 | 75.67 | 73.86 | **74.05** | 73.82 | 73.70 | **74.05** |
| haberman ($N = 306, n = 3$) | 66.86 | 67.21 | **69.19** | 68.17 | 67.20 | **73.83** | 73.80 | 73.20 | 73.18 | 73.80 |
| housing ($N = 506, n = 13$) | **76.24** | 75.73 | 75.37 | 75.57 | 75.73 | 91.65 | 91.70 | **92.68** | 92.65 | 91.70 |
| ILPD ($N = 583, n = 10$) | **74.01** | 73.66 | 73.56 | 73.77 | 73.66 | 71.11 | 71.07 | 71.68 | **71.79** | 71.07 |
| mammo. ($N = 830 \, n = 5$) | **87.73** | 87.72 | 87.70 | 87.68 | 87.71 | 81.00 | **81.20** | 80.99 | 80.94 | **81.20** |

Table 2: Average area under the curve (AUC) and correct classification rates (CCR) on UCI datasets ($m = 1$).

[Meta-Review · NeurIPS 2020]

This paper proposes a method for distributionally robust optimization under KL ambiguity sets for exponential families. Although KL ambiguity sets have their drawbacks, in particular not covering any changes in the inputs x, the present work produces a standard conic problem for a wide problem class via a novel analysis, provides good theoretical analysis, and yields good numerical results for a variety of small-scale classification problems. With the various clarifications that came up in the reviews, this paper makes a solid contribution to the DRO literature and will be quite welcome to the NeurIPS audience.